# Characterization of a pluripotent stem cell-derived matrix with powerful osteoregenerative capabilities

Eoin P. McNeill[1], Suzanne Zeitouni[1], Simin Pan[1], Andrew Haskell [1], Michael Cesarek [1], Daniel Tahan[1], Bret H. Clough[1], Ulf Krause [2], Lauren K. Dobson[3], Mayra Garcia[1], Christopher Kung [1], Qingguo Zhao[1], W. Brian Saunders[3], Fei Liu[1], Roland Kaunas[4✉] & Carl A. Gregory[1✉]

Approximately 10% of fractures will not heal without intervention. Current treatments can be marginally effective, costly, and some have adverse effects. A safe and manufacturable mimic of anabolic bone is the primary goal of bone engineering, but achieving this is challenging. Mesenchymal stem cells (MSCs), are excellent candidates for engineering bone, but lack reproducibility due to donor source and culture methodology. The need for a bioactive attachment substrate also hinders progress. Herein, we describe a highly osteogenic MSC line generated from induced pluripotent stem cells that generates high yields of an osteogenic cell-matrix (ihOCM) in vitro. In mice, the intrinsic osteogenic activity of ihOCM surpasses bone morphogenic protein 2 (BMP2) driving healing of calvarial defects in 4 weeks by a mechanism mediated in part by collagen VI and XII. We propose that ihOCM may represent an effective replacement for autograft and BMP products used commonly in bone tissue engineering.

[1] Department of Molecular and Cellular Medicine, Institute for Regenerative Medicine, Texas A&M Health Science Center, College Station, TX 77843, USA. [2] Institute for Transfusion Medicine and Cellular Medicine, University Hospital Muenster, Muenster, Germany. [3] Department of Small Animal Clinical Sciences, College of Veterinary Medicine and Biomedical Sciences, Texas A&M University, College Station, TX 77843, USA. [4] Department of Biomedical Engineering, Texas A&M University, College Station, TX 77843, USA. ✉email: kaunas@tamu.edu; cgregory@tamu.edu

Every year, approximately 600,000 fractures in the United States experience delayed or incomplete healing[1] at a cost of $200 billion[2–4]. Current therapies, while effective to varying degrees, have drawbacks. Bone morphogenetic protein-2 (BMP2) is effective, but it can have potentially life-threatening pro-inflammatory effects and cause ectopic bone formation[5–8]. Autologous bone graft remains the gold standard for bone repair, but limitations in availability of graft and the risk of donor site morbidity are significant concerns[3]. Synthetic and cadaveric bone products are abundant and low cost, but suffer limited bio-compatibility and batch variation[9,10].

A safe and manufacturable material that mimics anabolic bone is the primary goal of bone tissue engineering, but achieving this is challenging. Bone marrow (BM)-derived human mesenchymal stem cells (hMSCs) have emerged as a promising solution due to their osteoinductive and immunomodulatory properties[11,12]. Clinical trials demonstrate the safety of hMSCs[13] but donor variability and limited availability present roadblocks to translation[14]. In contrast, MSCs differentiated from induced pluripotent stem cells (iPSC) have the potential to overcome these issues as iPSCs represent a theoretically limitless and reproducible source of cells[15–19].

We have shown that extracellular matrix (ECM) secreted by osteogenically enhanced hMSCs (OEhMSCs) improves bone healing by providing an osteogenic microenvironment[20–22]. OEhMSCs were generated by exposure to the PPARγ inhibitor, GW9662, causing acceleration of canonical Wnt (cWnt) signaling by ablating inhibitory crosstalk from the PPARγ axis. The ECM, referred to as human osteogenic cell matrix (hOCM), contains collagens enriched in anabolic bone[20] including collagens VI and XII which promote secretion of osteogenic and angiogenic factors[22].

Herein, we report that hMSCs generated from an iPSC line (ihMSCs)[17] are highly osteogenic as compared to BM-hMSCs, and generate large amounts of osteogenic matrix (ihOCM) with potent osteogenic properties. In murine calvarial defect assays, ihOCM stimulates healing of bone without the need for co-administered cells and is several-fold more effective than BMP2 when administered at a standard effective dose. Transcriptional knockdown experiments indicate that the matrix functions in part through a mechanism involving attachment to collagens VI and XII. Collectively, these results suggest that ihOCM, manufactured from a robust and abundant source of cells, may represent an alternative to current bone regeneration agents such as bone graft or BMP2.

## Results

### Characterization of ihMSCs

The ihMSCs were generated from iPSCs by culture on Matrigel with TGFβ inhibition[17,23] resulting in fibroblastoid cells that proliferated readily on plastic while lacking expression of reprogramming factors.

The ihMSCs exhibited a spindle-shaped morphology typical of hMSCs[24] (Supplementary Fig. 1a) and proliferated at a rate comparable to hMSCs with an average doubling time of 20 h (Fig. 1a, b). The ihMSCs also generated colonies with efficiencies comparable to BM-hMSCs[24] (Supplementary Fig. 1b). Yields, colony-forming capacity, and doubling times remained stable until passage 4 but thereafter there was a reduction in yield (Fig. 1a), increase in doubling time (Fig. 1b), and reduced colony-forming potential (Supplementary Fig. 1b). The passage-dependent reduction in yield was most evident at higher cell densities encountered at the late log phase of each plating (Supplementary Fig. 1c). Cumulative yield began to deteriorate after day 70 (Fig. 1c), supporting the observation that the ihMSCs exhibited proliferative senescence and do not possess the same proliferative immortality

as the source iPSCs[25]. The ihMSCs also exhibited an immunophenotype typical of hMSCs[26] (Fig. 1d, Supplementary Fig. 2c). Like hMSCs, ihMSCS inhibited mixed lymphocyte expansion as measured by carboxyfluorescein-succinimidyl ester (CFSE) dilution assay (Supplementary Fig. 2a, b) and reduced output of TNFα by lipopolysaccharide-stimulated macrophages (Supplementary Fig. 2d). In assays of osteogenesis and chondrogenesis, ihMSCs generated mineralizing alizarin red S (ARS)-stained monolayers and toluidine blue-stained chondrogenic micromasses (Fig. 1e, Supplementary Fig. 3a), but only a modest number of adipocyte-like cells (Fig. 1e, Supplementary Fig. 3a). Addition of the PPARγ agonist, troglitazone, or the β-catenin inhibitor, CCT032374, individually (or in combination) to the adipogenic medium improved differentiation (Supplementary Fig. 4a, b). For comparison, two bone marrow-derived hMSCs (BM-hMSCs1 and 2) readily generated all three lineages (Fig. 1e, Supplementary Fig. 3a). Quantitative RT-PCR (qRT-PCR) for lineage-specific transcripts indicated that BM-hMSC1 was more osteogenic than BM-hMSC2, and ihMSCs exhibited superior osteogenic traits comparable to BM-hMSC1 (Fig. 1f, Supplementary Fig. 3b, c). Assays based on quantification of ARS-stained mineral[27] indicated the order of osteogenic capacity was ihMSC>BM-hMSC1>BM-hMSC2 (Fig. 1g). Osteogenic media contains dexamethasone to trigger mineralization by BM-hMSCs[27]. This was not necessary for ihMSCs, with mineralization occurring in the presence of only osteogenic supplements β-glycerophosphate and ascorbic acid (osteo-base media, OBM) (Fig. 1h). Collectively, these results demonstrate that ihMSCs possess the characteristics expected of MSCs, and exhibit superior osteogenic properties when compared to the BM-hMSCs.

### Osteogenic enhancement of ihMSCs

Osteogenic enhancement of BM-hMSCs with a PPARγ inhibitor, GW9662, occurs by release of negative crosstalk between the adipogenic PPARγ axis and osteogenic cWnt signaling[20,28]. During osteogenic stimulation, PPARγ is prevented from promoting glycogen-synthease-kinase-3b (GSK3β)-mediated proteolytic degradation of β-catenin and in turn, the stabilized β-catenin transcriptionally down-regulates PPARγ expression (Supplementary Fig. 5a)[29,30]. To explore the role of cWnt in ihMSC osteogenesis, monolayers were incubated with OBM for 7 days and subjected to subcellular fractionation and immunoblotting for major cWnt components. Compared to non-osteogenic standard culture medium, GSK3β was downregulated in the soluble cytosolic fraction and β-catenin was upregulated in the detergent-insoluble nuclear-containing fraction (Fig. 2a, b), indicative of upregulated cWnt signaling[31]. PPARγ isoforms were downregulated in the insoluble fraction, supporting the presence of negative crosstalk between the cWnt and PPARγ axes (Fig. 2a, b). When ihMSCs are incubated in OBM with GW9662, a further increase in β-catenin and down-regulation of GSK3β was observed (Fig. 2c, d), with increased activity of the osteogenic biomarker alkaline phosphatase (ALP) (Fig. 2e) and enhanced mineralization (Fig. 2f) even in the presence of potent levels of dexamethasone (Supplementary Fig. 5b). Collectively, the results indicated that GW9662 generates osteogenically enhanced ihMSCs (OEihMSCs) by an equivalent mechanism to that observed in BM-hMSCs[28].

OEihMSCs generated an abundance of ECM similar to the hOCM generated by OEhMSCs (hereafter referred to as ihOCM) but in larger quantities (Fig. 2g). Protein levels per unit mass of ihOCM were equivalent to hOCM generated from the BM-hMSCs and were unchanged by treatment with GW9662 (Fig. 2h). GW9662 treatment increased calcium levels in OEhMSC1 matrix (hOCM1) and ihOCM, but calcium remained relatively low in OEhMSC2 matrix (hOCM2) (Fig. 2i), further supporting the

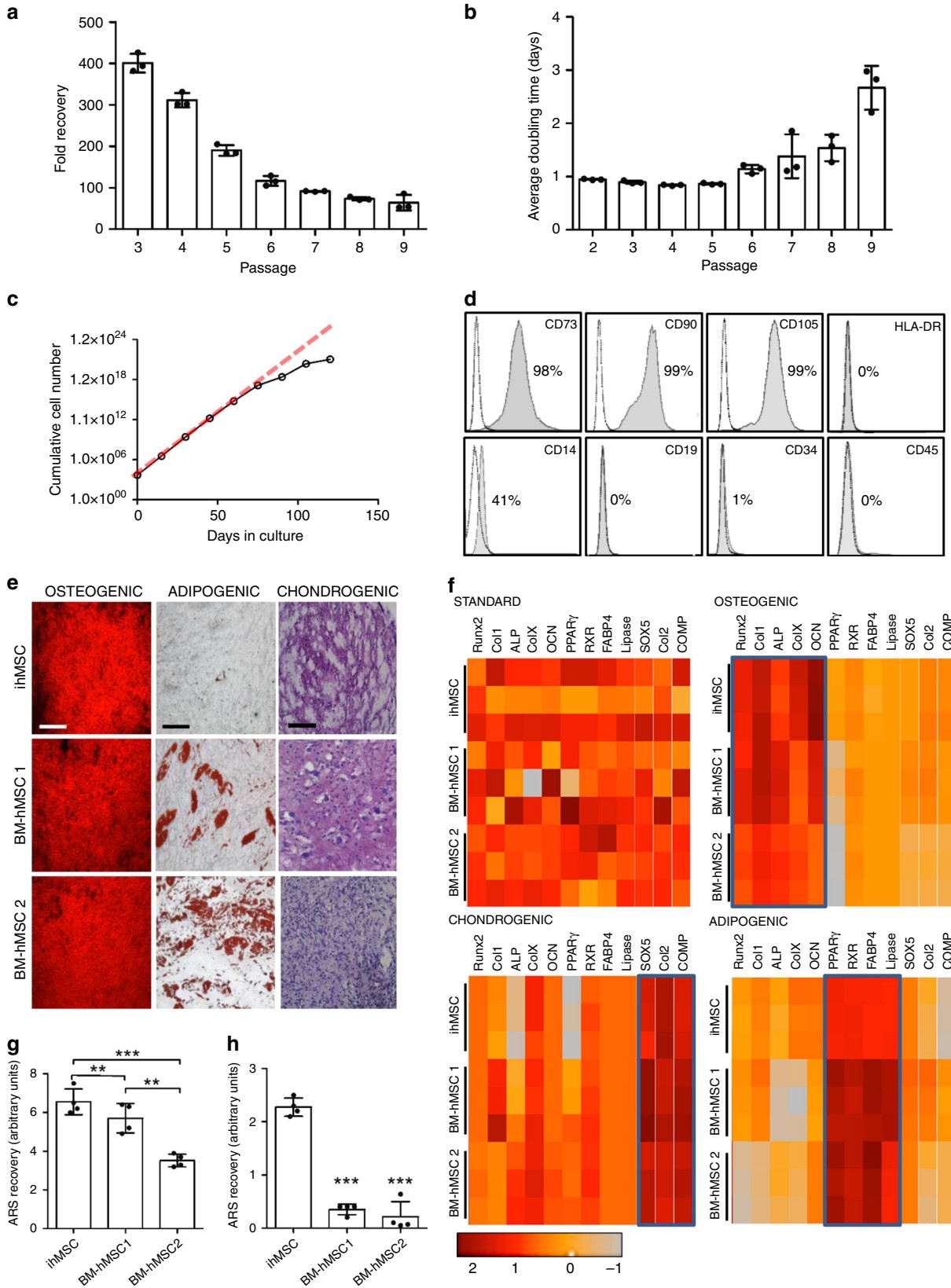

observation that OEihMSCs and OEhMSC1 cells were more osteogenic than OEhMSC2 cells. Glycosaminoglycan (GAG) levels were low but comparable in matrices generated from monolayers that received GW9662 (Fig. 2j). Collectively, these data indicated that GW9662 increased yields, but not basic composition of ihOCM when compared to BM-derived hOCM.

**Composition of the matrices**. To characterize the composition of the matrices, proteomic analysis was performed on preparations from GW9662-treated and untreated monolayers. Forty-nine polypeptides were identified with 16 shared between the matrices generated from untreated cells (Fig. 3a, Supplementary Table 2). GW9662 increased the number of polypeptides shared between

**Fig. 1 ihMSCs share many of the key characteristics of BM-hMSCs. a** Fold recovery in 7 days after plating at 500 cells per cm$^2$ after 3–9 passages. **b** Average doubling time for cultures in **a**. **c** Cumulative cell yield in 120 days. The red dashed line represents the initial rate of expansion. **d** Immunophenotype of ihMSCs. **e** ihMSCs differentiate into osteoblasts and chondrocytes but less so to adipocytes in standard assays. Bar = 100 μm for monolayers and 25 μm for chondrogenic assays. Each image is a representative of six images taken from three separate experiments. Controls presented in Supplementary Fig. 3a. **f** Heat map summarizing fold-change transcription of differentiation biomarkers. The scale represents z-score, raw data presented in Supplementary Fig. 2b. **g** ARS quantification assay comparing ihMSCs mineralization to two BM-hMSC preparations. **h** As **g**, but performed in the absence of dexamethasone. Statistics: panels **a**, **b**, **g**, **h** presented as means with SD, compared using one-way ANOVA with Tukey's or Dunnett's post-test. *$p < 0.05$, **$p < 0.01$, ***$p < 0.005$. For panels **a**–**f**, $n = 6$, for panels **i** and **j**, $n = 4$. Source data are provided as a Source Data file.

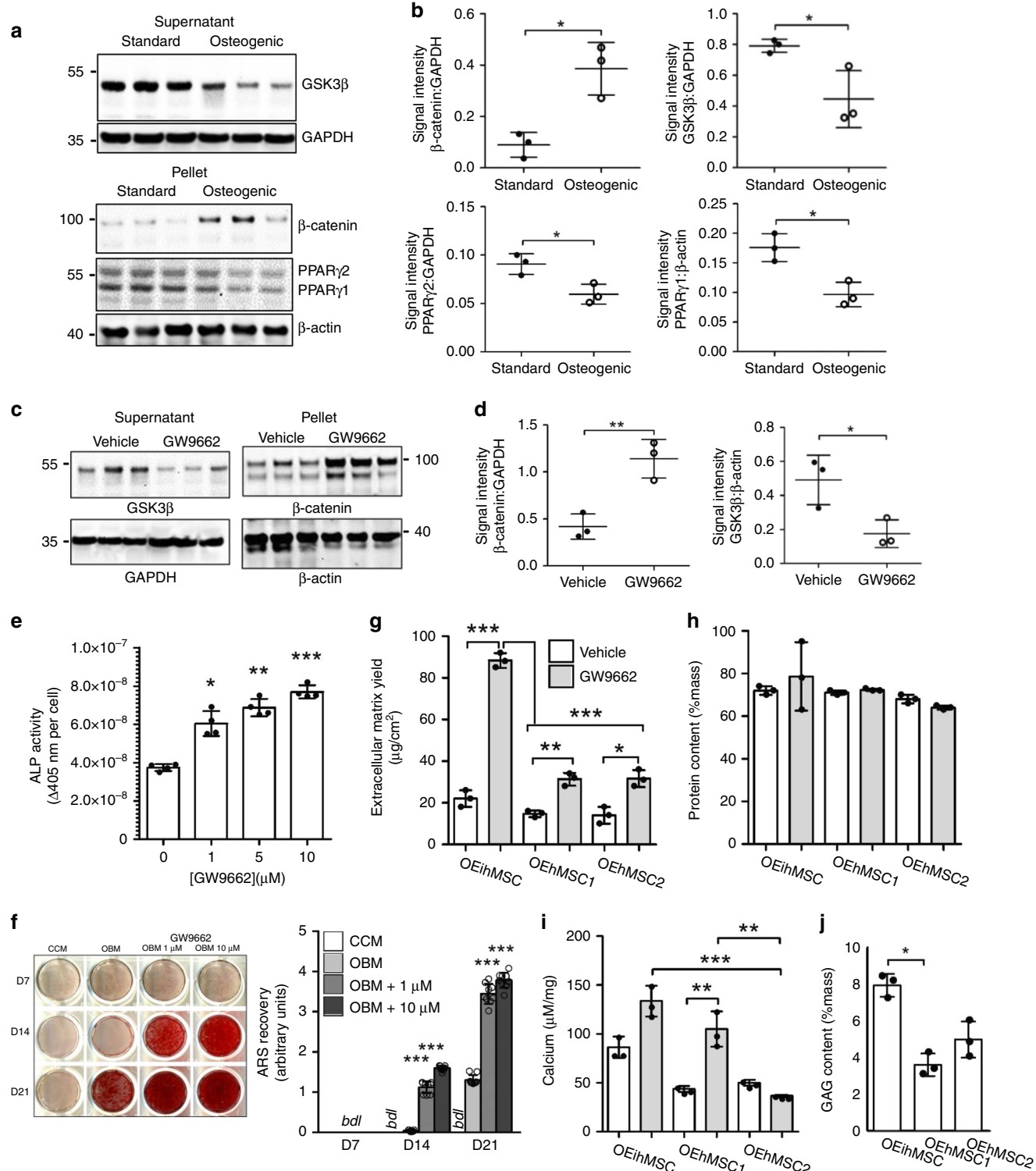

**Fig. 2 Osteogenesis by ihMSCs is regulated by the cWnt/PPARγ axis and PPARγ inhibition increases osteogenesis and matrix productivity. a** immunoblotting of ihMSC extracts (replicates of 3) for GSK3β, β-catenin, and PPARγ isoforms in the presence and absence of osteogenic stimulus. "Supernatant" and "Pellet" represent cytosolic and insoluble extracts respectively. **b** Densitometry measurements of images in **a**. **c** GW9662 exposure under osteogenic conditions causes further upregulation of insoluble β-catenin and downregulation of soluble GSK3β. **d** Densitometry measurements of images in **c**. **e** GW9662 exposure under osteogenic conditions causes upregulation of the osteogenic biomarker ALP in ihMSCs. **f** Culture conditions as **e**, but mineralization assays with ARS. Representative ARS-stained monolayers are presented (left) with quantification of staining (right). bdl below detectable limits. **g** Osteogenically enhanced ihMSCs on monolayers generate higher yields of ECM by mass when treated with GW9662. **h** Protein content in ECM from OEihMSCs and bone marrow-derived OEhMSCs. **i** Calcium content in ECM recovered from OEihMSCs and BM-OEhMSCs. **j** GAG content in ECM recovered from OEihMSCs and BM-OEhMSCs treated with GW9662. Statistics: For panels **b** and **d**, data presented with means and SD and analyzed by two-tailed Student's $t$-test. For panels **b**, **d**, **g–j**, $n = 3$; for **f**, $n = 5$; for **e**, $n = 4$. Source data are provided as a Source Data file. For panels **e–j**, data presented as means with SD, compared using one-way ANOVA with Tukey's or Dunnett's post-test. *$p < 0.05$, **$p < 0.01$, ***$p < 0.005$.

all matrices (Fig. 3a, Supplementary Table 3) at the expense of those shared between two of the preparations suggesting that GW9662 treatment has a normalizing effect on ECM composition from different cellular origins. Hierarchical clustering revealed a cell source-dependent relationship between treated and untreated matrices, but also indicated a strong similarity between hOCM1 and ihOCM (Fig. 3b), again reflecting the superior osteogenic capacities of ihMSCs and BM-hMSC1. Scanning electron microscopy (SEM) of matrices indicated the presence of nodules in ihOCM and hOCM1 (Fig. 3c) and electron diffraction spectrometry (EDS) confirmed that they contained calcium (Fig. 3d, Supplementary Fig. 6), but these were not readily detectable in hOCM2. Larger nodules were also detectable by phase-contrast microscopy on monolayers of OEihMSCs and OEhMSC1 that could be upregulated dose-dependently by GW9662 (Supplementary Fig. 7a, b). Given that OEihMSC1 and OEihMSCs exhibited the greatest osteogenic capacity, proteins shared in hOCM1 and ihOCM were examined. No proteins were exclusively shared by treated hOCM1 and ihOCM, but eight polypeptides were shared by untreated matrices originating from the α1- and α2-chains of collagen VI, α1-collagen XII, integrin β1, and collagen IV. Quantitative immunoblotting further indicated that collagen VI and XII synthesis was highest in OEihMSCs followed by OEhMSC1, then OEhMSC2 (Fig. 3e), correlating with their observed in vitro osteogenic capacities. Additional proteomic analysis on electrophoretically fractionated ihOCM confirmed the presence of collagen VI and XII, as well as periostin and TGFβ-IPig-h3, known to form associations with collagen VI and XII[32–36] (Fig. 3f). Increased transcription of a range of collagens associated with anabolic osteoid was detectable in response to GW9662 as compared to vehicle, including collagens VI and XII (Supplementary Fig. 7c). Furthermore, treated ihOCM upregulated the activity of ALP and the early osteogenic biomarker osteoprotegerin (OPG) by attached ihMSCs and hMSCs more effectively than untreated ihOCM (Fig. 3g, hMSCs shown), suggesting that the enhanced levels of collagen VI, collagen XII upregulated by GW9662 might represent the primary osteogenic stimuli in the matrices. Assays of late-stage osteogenic biomarkers were prevented by competition from cell-derived ECM that accumulated during long-term culture. Osteogenic responses elicited by ihOCM could only be detected in the presence of OBM and were marginal in standard culture medium.

**Collagen VI and XII enhance proliferative and osteogenic characteristics of hMSCs.** Given the importance of type VI and type XII collagen in stimulating osteogenesis[37] and their role in bone development[20,38–40], we permanently knocked down transcription of collagen VI or collagen XII in OEhMSC1 cells (Fig. 4a–c). Interestingly, knockdown of collagen VI (KD6) caused transcriptional downregulation of collagen XII, whereas knockdown of collagen XII (KD12) resulted in upregulation of collagen VIα3, suggesting a regulatory relationship between the

genes (Fig. 4b, c). KD6 and KD12 cells exhibited reduced proliferation (Fig. 4d, Supplementary Fig. 8b) and a reduction in colony forming potential (Fig. 4e, f, Supplementary Fig. 8d). Morphology was affected in both cases, resulting in the development of long cellular processes and a reduced tendency to form intercellular contacts (Supplementary Fig. 8a, e). Apoptosis levels in scrambled, KD6 and KD12 cultures were very low, and were not affected by knockdown of either collagen (Supplementary Fig. 8c). Immunophenotype and the capacity to generate chondrocytes were also unaffected (Supplementary Figs. 8f and 9a), but KD6 cells had reduced capacity to generate adipocytes (Supplementary Fig. 9b). KD6 and KD12 cells had reduced capacity to form mineralized monolayers (Fig. 4g, h) and also reduced ALP activity and OPG output (Fig. 4i). Immunocytochemistry for collagen VI and XII in scrambled cells revealed even distribution of collagen XII with additional punctae of co-localized VI and XII in approximately 60% of the intercellular contacts (Fig. 4j, top right, closed arrows). Blockade of collagen XII resulted in re-distribution of collagen VI punctae within the cell body rather than at contacts (Fig. 4j, below left, asterixes). In KD6 and KD12 cells, intercellular contacts were evident, but less common (Fig. 4j, lower, open arrows). Taken together with the observed transcriptional co-regulation, these results suggest a functional relationship between the two collagens relating to intercellular contact.

**Collagen VI and XII knockdown diminish the bone healing potency of hMSCs.** To evaluate the in vivo osteogenic potential of the KD hMSCs, the cells were implanted into calvarial defects in nude mice. After 4 weeks, micro-computed tomography (μCT) was employed to determine the healing index (HI) where the value of 1 was defined as volumetric bone deposition equivalent to the contralateral side whereas 0 was defined as no healing[20]. Both KD6 and KD12 cells had markedly reduced capacity to heal defects, whereas significant healing occurred in defects that received scrambled hMSCs (Fig. 5a–c). Histology of the healing calvarial bones confirmed the μCT data (Fig. 5a, b). These data demonstrate that both collagen XII and VI are necessary for osteogenic healing by MSCs.

**The bone healing properties of ihOCM.** When co-administered with OEhMSCs or human bone marrow (hBM) cells, BM-derived hOCM improves the osteoregenerative capacity of the cells through improved cell retention and osteogenic enhancement[20,22,37]. To explore the osteogenic efficacy of ihOCM, OEhMSCs or hBM was co-administered with ihOCM or gelatin foam (GF) in the calvarial defect model. Negative controls received no treatment, and positive controls received 0.1 mg mL$^{-1}$ BMP2 administered with GF, a dose that exhibits osteoregenerative activity[41,42]. In control defects and defects that received cells on GF, a modest HI was observed with the best bone growth occurring with GF and OEihMSCs and the

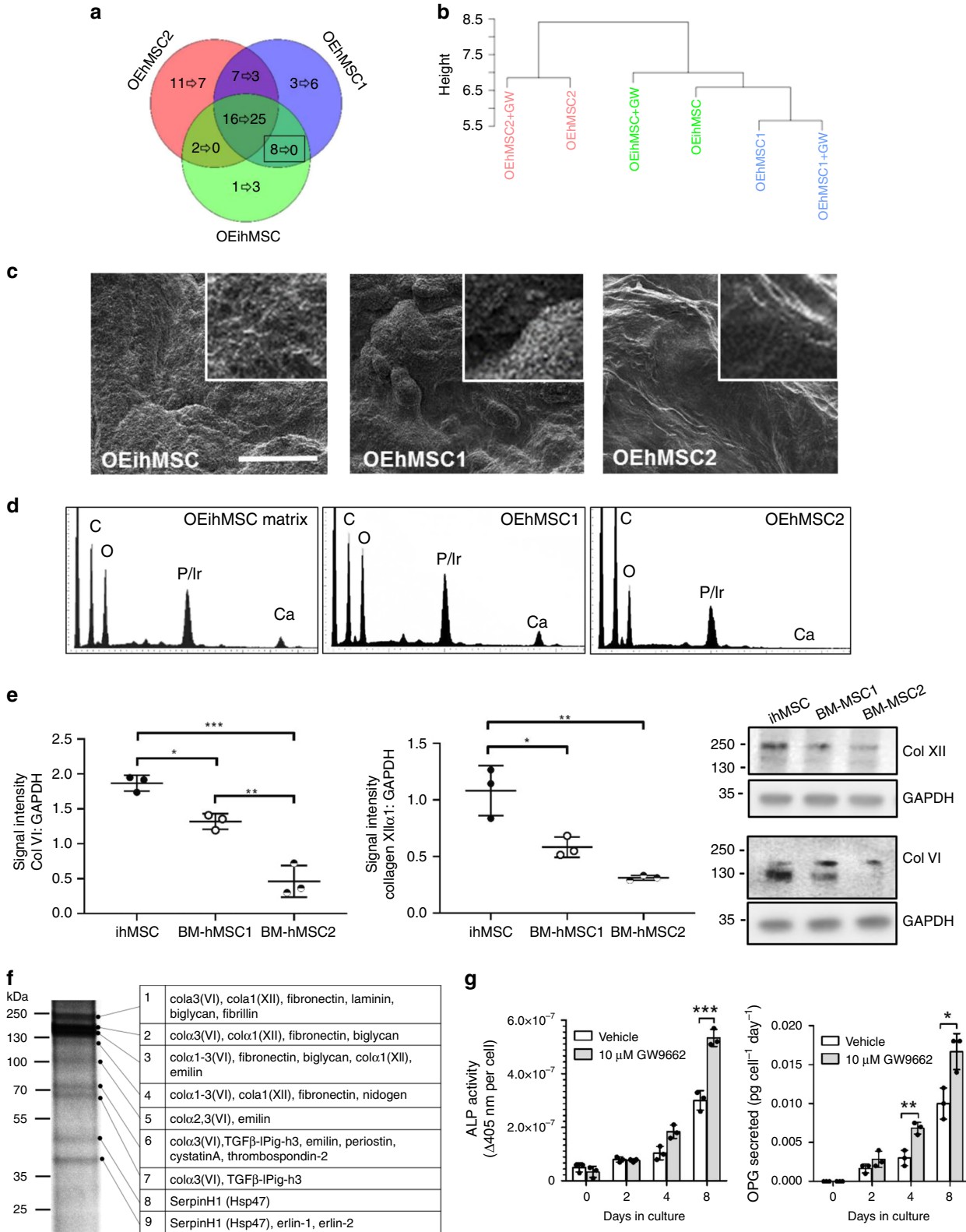

BMP2 control (Fig. 6a, c, Supplementary Fig. 10b–d, ST4). BMP2 induced a variable healing response that was only superior to negative controls (Fig. 6a, c, Supplementary Fig. 10c, d, ST4), and inspection of histological sections revealed the formation of diffuse patches of immature bone (Supplementary Fig. 10c, arrowed). In every defect that received ihOCM, healing was far more pronounced, with complete healing (HI > 1) evident in all but two cases

(Fig. 6a–d, Supplementary Fig. 11a, ST4). Of significant note was the observation that the most bone growth occurred in defects that received no additional cells (Fig. 6a–c) (HI 2–3) with the least bone formed in defects that received OEihMSCs (HI 0.5–1.8) (Fig. 6a–c). Bone surface:volume ratios indicated that the de novo bone was more compact in defects that received ihOCM (Fig. 6d, ST5) but the bone mineral density (BMD) was equivalent between experimental

**Fig. 3 ECM purified from monolayers of OEihMSCs and BM-OEhMSCs contains enriched levels of collagen VI and XII and can accelerate expression of osteogenic biomarkers. a** Venn diagram illustrating the distributions of different protein signatures detected in matrix extracted from OEihMSCs and two BM-OEhMSCs preparations. Numbering represents individual proteins detected without (before arrow) or with (after arrow) GW9662 treatment. Identities of proteins shared by each MSC preparation are provided in Supplementary Tables S2 and 3. Boxed number represents matrix proteins shared by OEihMSCs and highly osteogenic donor OEhMSC1, comprising collagen VI, collagen XII, and integrin receptor fragments: hierarchical clustering of proteomic data indicates that OEihMSC ECM clusters with ECM generated from the highly osteogenic OEhMSC1 donor. **c** SEM of matrices purified from OEihMSCs and OEhMSC monolayers revealing an abundance of dense vesicles on OEihMSC and OEhMSC1 matrix. Each image is a representative of three images taken from three separate specimens. **d** EDS measurements confirm the presence of calcium in the osteogenic OEihMSC and OEhMSC1 matrix, but not in the poorly osteogenic OEhMSC2 matrix. Iridium from sample processing masked phosphate determinations. Raw EDS data provided in Supplementary Fig. 6. **e** Densitometric immunoblotting for type VI and type XII collagen demonstrates higher expression in OEihMSCs as compared to BM-OEhMSC preparations. **f** Proteomic analysis of matrix from OEihMSCs with greater electrophoretic separation confirmed the presence of collagen VI, XII, and their accessory proteins periostin and TGFβ-IP-igh3. **g** ihMSCs were attached to ihOCM generated in the presence of GW9662 (gray) or vehicle (white) and subjected to osteogenic stimulus. ALP activity (left) and OPG secretion (right) were measured during the period of osteogenic differentiation. Statistics: For panels **e**–**j**, data presented as means with SD and analyzed using one-way ANOVA with Tukey's or Dunnett's post-test. *$p < 0.05$, **$p < 0.01$, ***$p < 0.005$. For panels **e** and **g**, $n = 3$. Source data are provided as a Source Data file.

groups (Supplementary Fig. 11b). When the BMD of the de novo bone in defects was compared to mature calvarial, femoral, vertebral, and iliac bone of the host animals, densities were superior to pelvic bone, slightly lower than femoral and vertebral bone and equivalent to calvarial bone (Supplementary Fig. 11c). Similar measurements were made for surface:volume ratios, and the de novo bone was found to be more compact than the mature bone specimens (Supplementary Fig. 11d). Collectively, these radio-morphometric parameters predict that the de novo bone does not differ substantially from mature bone tissue in terms of strength and durability.

Thresholding studies confirmed that ihOCM was radiolucent as compared to mature bone and could be effectively excluded from scans (Supplementary Fig. 10a). Histological analysis confirmed that the de novo bone exhibited a dense trabecular morphology but with isolated patches of fibrous material that was subject to intense remodeling by osteoclasts (Fig 6a, b, Supplementary Fig. 11e). Interestingly, the ihOCM-only defects that exhibited the greatest degree of healing had the most osteoclast activity when stained for tartrate resistant acid phosphatase (TRAP) (Supplementary Fig. 11e).

## Discussion

Products for bone repair can be categorized as processed products such as demineralized bone matrix (DBM), synthetic products such as hydroxyapatite or tricalcium phosphate, biologicals such as BMP2, or cellular allografts that combine live human cells with bone tissue. While these products utilize different approaches to address a challenging clinical problem, each have significant drawbacks. DBM and allograft products suffer from batch variation and the risk of pathogen transmission[9,43]. Synthetic products are cost-effective, but suffer from poor biocompatibility and cytotoxicity issues[44]. BMPs are osteoinductive, but they have exhibited life-threatening side effects such as excessive ossification, radiculopathy, and inflammation[45]. Despite decades of development of bone mimetic materials and osteoinductive agents, the gold standard for orthopedic repair continues to be autograft, but availability is limited and donor site morbidity is a concern[9,10].

A safe cell-based product such as hMSCs represents an attractive solution to the problems associated with bone repair technologies, but as we have learned with DBM, donor-derived materials suffer from batch variation and the potential for pathogen transfer. The use of iPSCs as a reproducible and donor-free source of progenitor cells circumvents these concerns, providing a theoretically infinite source of cells with predictable qualities. To this end, we generated ihMSCs from a single line of

iPSCs[17]. Unlike iPSCs, the ihMSCs exhibited proliferative senescence (Fig. 1c) and were not tumorigenic[41]. The ihMSCs therefore represent the best of both worlds, originating from a theoretically infinite source of genetically identical iPSCs, but becoming non-tumorigenic upon differentiation. Thereafter, ihOCM processing further contributes to safety by destroying nucleic acids and inactivating pathogens by solvent treatment.

In agreement with other studies[23,46–49], we observed that ihMSCs had an immunophenotype similar to hMSCs and exhibited trilineage differentiation potential, but with limited adipogenic capacity. Standard in vitro assays demonstrated that the ihMSCs were uniquely suited to osteogenic differentiation, even in conditions that lacked dexamethasone which is necessary for BM-hMSCs[27]. The osteogenic capacity of ihMSCs has been reported in several studies[47,50,51], but the ihMSCs described herein were superior to osteogenic BM-hMSC preparations.

In mesenchymal progenitors such as MSCs, adipogenic and osteogenic axes are homeostatically regulated by an inhibitory relationship driven by PPARγ-mediated degradation of β-catenin and cWnt-mediated inhibition of PPARγ expression (Supplementary Fig. 5)[29,30]. As with BM-hMSCs, ihMSCs upregulate cWnt signaling in response to osteogenic stimuli, accompanied by inhibition of the adipogenic PPARγ axis and stimulation of osteogenic biomarkers. Like BM-hMSCs, further upregulation of cWnt and concomitant osteogenesis can be achieved if PPARγ is inhibited by GW9662 (refs. [20,28]). Coupled with the observation that β-catenin inhibition increases adipogenic differentiation in ihMSCs, the results indicate that the inversely regulated osteogenic and adipogenic mechanism driven by cWnt and PPARγ is present in ihMSCs.

When ihMSCs are treated with GW9662 under osteogenic conditions, they adopt an OEihMSC phenotype accompanied by the secretion of a dense ECM, referred to as ihOCM, which stimulates the osteogenesis of attached osteoprogenitors, and is rich in calcified nodules and the osteogenic collagens VI and XII. We show here that collagen VI and XII levels correlate with the osteogenic potential of hMSCs and ihMSCs. Collagen VI and XII is enriched in embryonic osteoid[52,53] and regulates the morphology and polarity of osteoblasts in developing mice[39,40]. When hOCM-resident collagen VI and XII is blocked, attached hMSCs reduce output of osteogenic factors such as OPG and BMP2 (ref. [22]) suggesting that the collagens have the capacity to transduce an osteogenic signal. While the mechanism of collagen VI and XII-induced osteomodulation is not well defined, it is known that both collagens co-localize to contacts between osteoblasts that have a stimulatory effect on osteogenesis in vitro[38]. Indeed, similar structures were observed in our OEhMSCs with punctae of VI and XII co-localized to about half of intercellular contacts.

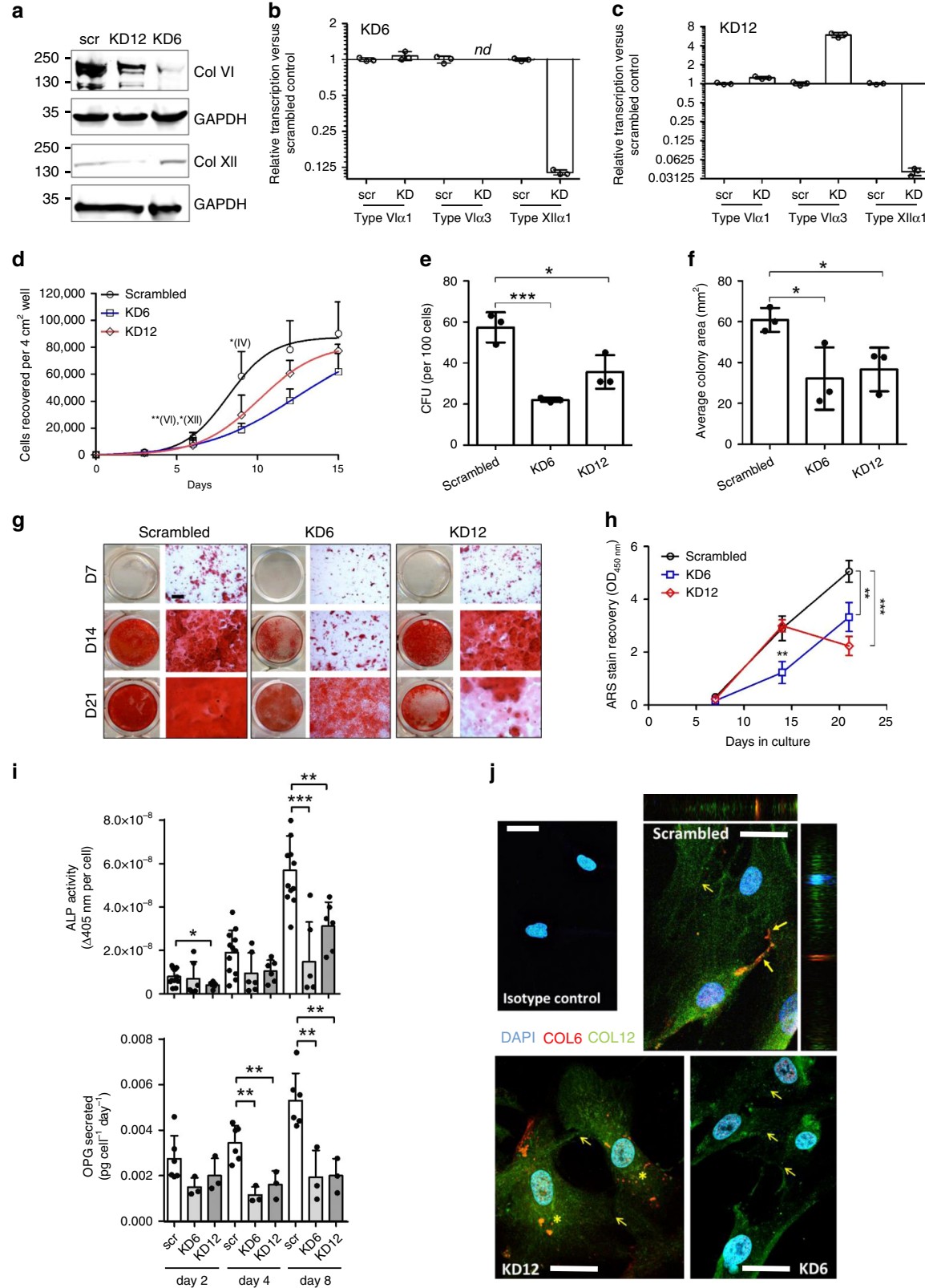

Interestingly, blockade of collagen XII resulted in a random redistribution of type VI collagen punctae, suggesting that type XII might regulate the distribution of type VI collagen. These putative co-localized structures appear to play a key role in hMSC-mediated osteogenesis because blockade of collagen VI or XII caused loss of proliferative capacity and a catastrophic loss of

osteogenic potential. Knockdown studies revealed the existence of a transcriptional feedback loop where blockade of collagen XII causes upregulation of collagen VI which, in turn, downregulates type XII transcription. This observation, in addition to evidence of colocalization, posits a functional relationship between collagen VI and XII that probably involves physical interaction.

**Fig. 4 Knockdown of collagen VI or collagen XII transcription results in viability, proliferative, and osteogenic deficiencies in vitro. a** Lysates of hMSCs harboring constructs expressing scrambled siRNA (scr), siRNA directed against collagen type VIα3 transcript (KD6), or collagen type XIIα1 transcript (KD12) were blotted and probed with antibodies against type VI collagen (Col VI) or type XII collagen (Col XII). Blots were performed independently three times. **b** Transcription of collagen VIα1, VIα3, and XIIαl in KD6 hMSCs measured by qRT-PCR. Results expressed as fold change compared to scr hMSCs. nd = signal not detected. **c** As panel **b**, but for KD12 hMSCs. **d** Growth curves of hMSCs over 15 days after an original seeding of 100 cells per $cm^2$. **e** Single cell-derived colonies generated per 100 plated cells. **f** Average colony area. **g** Monolayer mineralization determined by ARS staining for calcium deposits after 7–21 days of exposure to basal osteogenic media supplemented with dexamethasone. Entire wells (left in each data set) or micrographs (right in each data set) are presented (bar = 100 μm). **h** Quantification of ARS determined by extraction and spectrophotometric quantification. **i** ALP activity (above) of intact monolayers after 8 days of exposure to basal osteogenic media. Rate of OPG secretion (below) after 4 days of exposure to basal osteogenic media. **j** KD6, KD12, or scrambled hMSCs were probed with anti-collagen VI (red) and anti-collagen XII (green) and stained for nuclei with DAPI (blue). Intercellular contacts harboring yellow punctae of co-localized collagen VI and XII (closed arrows) are detectable in scrambled cultures but not in the case of KD6 or KD12 even though contacts are evident (open arrows). In KD12 cells, punctae of collagen VI are randomly distributed throughout cell bodies. In the case of the scrambled micrograph, orthogonal images are provided. Representative imaging from three independent experiments. Bar = 20 μm. Statistics: Data are presented with means and SD (error bars) and analyzed using one-way ANOVA with Tukey's or Dunnett's post-test. *$p < 0.05$, **$p < 0.01$, ***$p < 0.005$. Panels **b**–**f** and **h**, $n = 3$, for panel **i** $n = 6$, or $n = 12$ for scr. Source data are provided as a Source Data file.

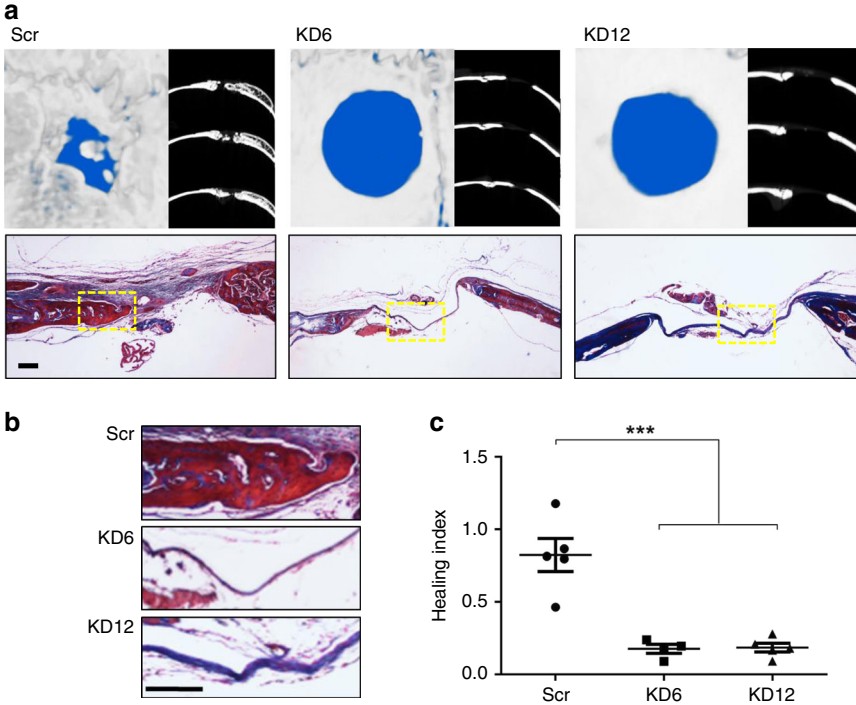

**Fig. 5 Knock down of collagen VI or collagen XII transcription results in osteogenic deficiencies by OEhMSCs in vivo.** Immune-compromised nude mice received a 4 mm diameter, full-thickness circular calvarial defect followed by $2 \times 10^6$ scrambled, KD6, or KD12 OEhMSCs delivered in clotted plasma. After 4 weeks, calvaria were scanned by μCT and subjected to histology. **a** μCT reconstructions (above left), axial cross-sections (above right), and Masson's trichrome-stained sections of calvarial specimens after 4 weeks of healing (bar = 250 μm). **b** High-power micrographs of healed specimens indicating trabecular micro-structure of newly formed bone (bar = 75 μm). Yellow boxes in panel **a** indicate the enlarged region in panel **b**. **c** Healing index of defects. Image data representative of five pictures generated from each independent specimen. Statistics: data are presented with means (horizontal line) with SD (error bars). The data were compared using ANOVA with Tukey's post-test. ***$p < 0.005$, $n = 5$ except KD6 where $n = 4$. Source data are provided as a Source Data file.

There is precedent for this hypothesis in the literature, because an ihOCM component and early osteogenic marker[33] TGFβ-IPig-h3 interacts with both collagens[32,34,35]. When bound to collagen XII, TGFβ-IPig-h3 activates focal adhesion kinase[35], whereas collagen VI has the capacity to signal independently via nerve/glial antigen 2 (ref. [54]), simultaneously triggering attachment, survival, and osteogenic pathways. In humans, a functional relationship between collagen VI and XII is further suggested by the observation that mutations in either collagens cause myopathies and forms of Ehlers–Danlos syndrome[55].

The function of OCM can be attributed to the action of collagen VI and XII, but this does not definitively explain the superiority of the GW9662-treated ihOCM when compared to

BM-hMSC-derived counterparts. Of the three peptide signatures exclusive to GW9662-ihOCM and not present in any of the GW9662-untreated matrices, fibulin-2 was detected. Interestingly, fibulin-2 possesses dual affinities for collagen IV and collagen VI, and compared to other ECM components, collagen IV has strong affinity for BMP2 while potentiating its activity suggesting that it could serve to link BMP:collagen VI complexes to ihOCM via collagen VI[56].

The calvarial defect model is a suitable in vivo initial model to test the osteogenic capabilities of OCM because it can be readily performed in immune-compromised rodents, will not spontaneously heal[28], requires no additional fixation, and does not require additional scaffolds to satisfy stiffness requirements or

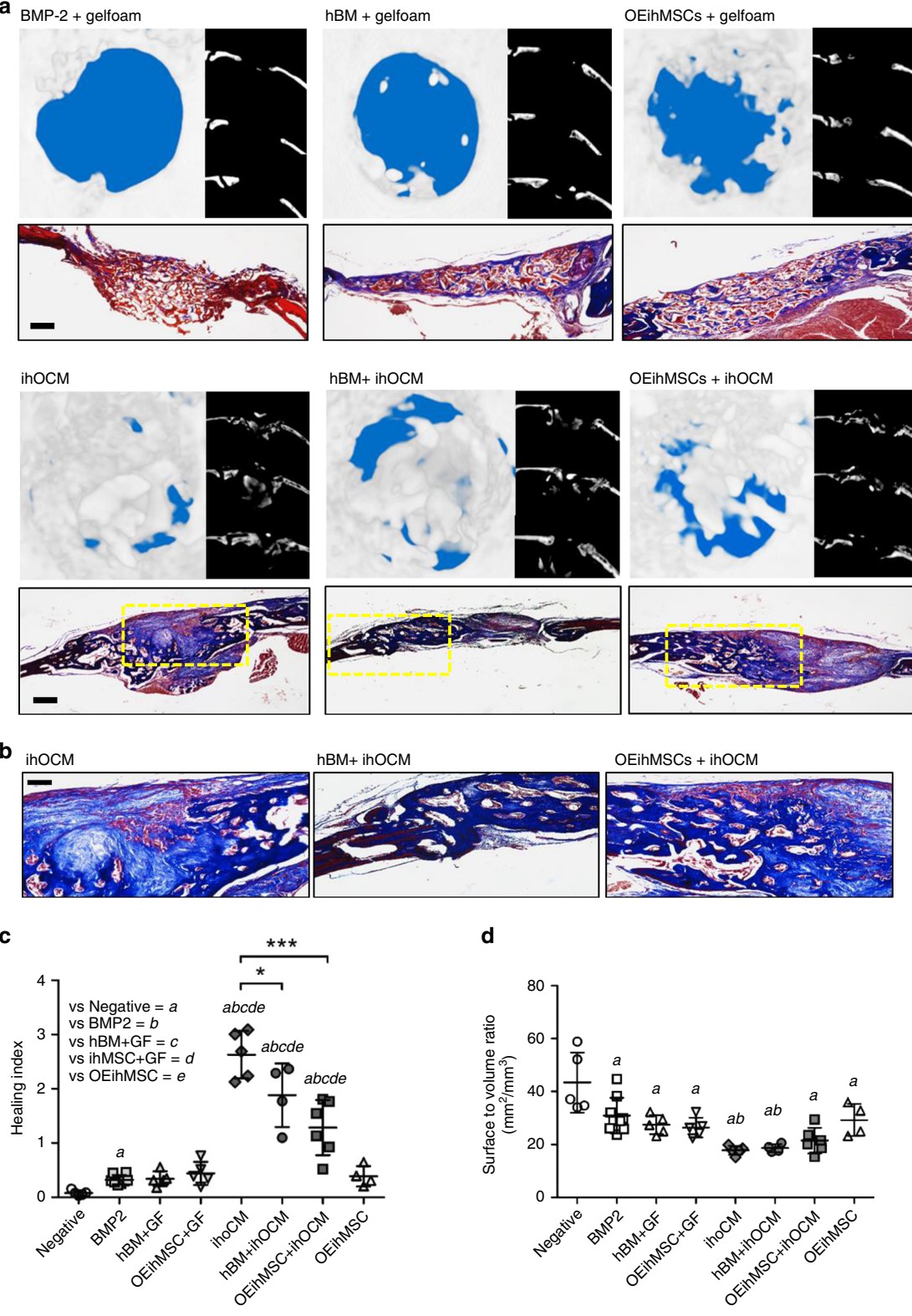

immobilization. The assays are relatively rapid to perform (4 weeks) and endpoint analyses are straightforward and readily interpretable. The ihOCM can also be tested in its pure, native form, using mm$^3$ quantities. We have previously demonstrated that hOCM derived from BM-OEhMSCs stimulates osteogenesis by attached OEhMSCs through extended cell attachment, support of survival, and stimulation of secretion of osteogenic factors[20–22,57]. When ihOCM was co-administered with OEihMSCs or hBM to

calvarial defects, an extremely high level of healing was observed after 4 weeks, with significantly greater healing than defects receiving a regenerative dose of BMP2. Surprisingly, the level of healing was further improved by omission of exogenously added cells, contrasting with OEhMSC-derived hOCM, which exhibits limited intrinsic efficacy without a co-administered progenitor cell[20,22]. While the calvarial healing model offers several advantages, it is challenging to perform biomechanical measurements on

**Fig. 6 ihOCM exhibits enhanced osteoregenerative capacity in the presence or absence of exogenously added osteo-progenitor cells.** Immune-compromised nude mice received a 4 mm diameter, full-thickness circular calvarial defect. Gelatin foam (GF) or ihOCM (ihM) was inserted into the defect and it was allowed to heal alone or in the presence of BMP2, OEihMSCs, or hBM cells. After 4 weeks, mice were euthanized and calvarial bones were dissected out for analysis. **a** μCT reconstructions (above left), axial cross-sections (above right), and Masson's trichrome-stained sections of calvarial specimens after 4 weeks of healing (bar = 250 μm). **b** High-power micrographs of healed specimens indicating trabecular micro-structure of newly formed bone (bar = 75 μm). Yellow boxes in panel **a** indicate the enlarged region in panel **b**. **c** Healing index of defects. Statistical comparisons of ihOCM specimens compared to controls are annotated **a–e** and cross-comparisons between ihOCM-treated specimens are denoted by lines and asterisks. **d** As panel **c**, but the surface to volume ratio is plotted. Image data representative of five pictures generated from each independent specimen. Statistics: Data are presented with means (horizontal line) with SD (error bars). The data were compared using one-sided ANOVA with Tukey's post-test. ***$p < 0.005$. For all groups $n = 5$, except ihM+hBM $n = 4$, ihM+OEihMSC $n = 6$, and BMP2 $n = 10$. Source data are provided as a Source Data file.

the defects. As an alternative, we employed radiomorphometric predictors of bone strength and durability, BMD and surface: volume ratio[58]. De novo bone generated in the presence of ihOCM was compared to calvarial, iliac, vertebral, and femoral bones from the same host animals. While slightly lower than the BMD of femoral and vertebral bone, the BMD of the newly healed bone was equivalent to calvarial bone and was greater than the BMD of iliac bone tissue. Surface:volume ratios were lower in the ihOCM generated bone than all other tissues measured, suggesting that the tissue is more compact. Collectively, the results predict that the bone formed in the presence of ihOCM has strength and durability within the range of homeostatic bone tissue from several sources.

In the present study, BMP2 demonstrated a variable level of efficacy despite being administered at an active dose in mice[41,42]. This finding is likely due to the short 4-week duration of the assay and conservative thresholding of the μCT. Most calvarial healing assays are performed for 6–8 weeks[59] and while BMP2-stimulated bone is generally robust after remodeling, the immature osteoid is difficult to detect radiologically[60,61]. Accordingly, histological analysis confirmed the presence of diffuse clusters of immature bone in defects that received BMP2 (Supplementary Fig. 10c), but this material was not robustly detected by μCT.

Exogenously added OEihMSCs or hBM reduced the capacity of ihOCM to heal the calvarial defects. The reason for this surprising observation is unclear, but healing was correlated with the presence of osteoclasts at the defect, and it is feasible that addition of large numbers of osteoblast progenitors could disrupt homeostasis necessary to simultaneously orchestrate remodeling and bone deposition. The results challenge the dogma that large doses of osteogenic cells are supportive of healing and suggest that ihOCM alone, when administered to an adequately perfused injury site, is sufficient to drive repair.

MSCs derived from iPS cells have been described by several groups and their potential as a cytotherapeutic agent for bone repair has been reported[18,19,47,62]. These studies have yielded promising results, but also concern related to biological variation between iPSC lines. While the poorly understood nuances of iPSC variation raise interesting mechanistic questions, they can also serve as a roadblock to autologous therapeutic approaches. Instead, this work offers the means to generate ihOCM with very reproducible composition from a single iPSC line. Because the starting material and final product are characterized and cell free, it could be utilized in the allogeneic setting without the concern of donor variation.

The data presented herein demonstrate that ihOCM exhibits a level of efficacy that challenges current approaches to bone repair. In its ECM form, ihOCM is unlikely to diffuse from the site of administration and does not appear to induce production of supraphysiological levels of bioactive factors. Furthermore, purified ihOCM does not contain cells, pathogens, or nucleic acids. While careful work is necessary to demonstrate safety and efficacy in large animal calvarial and long-bone models as well as human

recipients, these attributes support the probability that ihOCM is a safe and translatable improvement of current bone repair technologies.

## Methods

**Tissue culture.** Protocols related to the recovery and use of human biomaterials were approved by the Texas A&M Health Science Center Institutional Review Board. Human induced pluripotent stem cell-derived mesenchymal stem cells (ihMSCs) were generated by Zhao et al.[17]. hMSCs were acquired from the Texas A&M Health Science Center Institute for Regenerative Medicine MSC distribution facility in accordance with institutionally approved protocols. Both ihMSCs and hMSCs were cultured as previously described[63] in complete culture medium (CCM) consisting of alpha-minimal-essential-medium (αMEM; Life Technologies, Carlsbad, CA) supplemented with 20% (v/v) fetal bovine serum (FBS; Atlanta Biologicals, Norcross, GA), 2 mM L-glutamine (Life Technologies) and 100 U mL$^{-1}$ penicillin and 100 μg mL$^{-1}$ streptomycin (Life Technologies). For expansion, cells were seeded at 100 cells per cm$^2$ and media was replaced every 2 days. Adherent cells were recovered using 0.25% (w/v) trypsin/ethylene diamine tetra-acetic acid (EDTA, Life Technologies) when a density of about 10,000–15,000 cells per cm$^2$ (70–80% confluency) was reached (8–9 population doublings). Cells were cryopreserved in αMEM supplemented with 50% (v/v) FBS and 5% (v/v) DMSO (Sigma, St. Louis, MO). MSCs at passages 2–4 were used for experiments unless otherwise stated. Micrographs were obtained using an inverted microscope (Nikon Eclipse TE200) with a digital camera attached (Nikon DXM1200F). Unless otherwise stated, experiments were performed with cells up to passage 4, where passage is defined as 8–9 population doublings at sub-confluence.

**Colony assays.** One hundred MSCs were plated in a 152 cm$^2$ tissue culture plate (Corning) in the presence of CCM. After 3 weeks, the media was aspirated, plates were washed with PBS, then stained for 10 min in crystal violet suspended in 50% (v/v) methanol (Fisher Lifesciences). Plates were washed in excess dH$_2$O and colonies were counted and recorded as a percentage. In some cases, plates were digitally scanned and subjected to image analysis using CTAn software (Bruker, Contich, Belgium).

**Immunophenotyping.** MSCs were recovered with trypsin/EDTA and resuspended in phosphate-buffered saline (PBS; Life Technologies) supplemented with 2% (v/v) FBS (Atlanta Biologicals) with fluorophore-tagged antibodies (Becton Dickinson, Franklin Lakes, NJ or Beckman Coulter) for 30 min on ice. All antibodies were used at a 1:200 dilution. Antibodies against CD11b (clone BEAR1, #IM0530U), CD14 (RMO52, #IM0650U), CD19 (J3-119, #IM1285U), CD34 (581, #IM1871U), CD45 (J.33, #IM0782U), CD73 (AD2, #B68176), CD79a (HM47, #IM2221U), CD90 (Thy-1/310, #IM1839U), CD105 (TEA3, #B76299), and HLA-DP, DQ, DR (9-49, #18142), or appropriate isotype controls (FITC: B76299, 6603864, 6603853, 6603855, B36627; PE: A09141, IM0670U) were used. The cells were analyzed using a Cytomics FC500 flow cytometer (Beckman Coulter) and data were processed using the manufacturer's software (CXP) (for gating strategy, see Supplementary Fig S2c).

**Lymphocyte activation assays.** Assays were performed based on CFSE dilution[21,64]. Briefly, human allogeneic peripheral blood mononuclear cells (PBMC) were labeled with 2.5 μM carboxyfluorescein diacetate, succinimidyl ester (CFSE; Invitrogen, Carlsbad, CA). PBSCs (25,000) from two immunologically mismatched donors were co-cultured with 12500−250 OEihMSCs (or hMSCs) in RPMI 1640 medium (Thermo Fisher) supplemented with 10% (v/v) human AB serum (CellGro, Corning, NY) and antibiotics for 7 days. Cultures were analyzed on a Cytomics FC500 flow cytometer (Beckman Coulter) and data were processed using the manufacturers' CXP software (for gating strategy, see Fig S2b).

**Co-culture of ihMSCs and macrophages.** RAW 264.7 murine macrophage-like cells (American Type Culture Collection) were labeled with cytosolic green fluorescent protein (GFP) using a commercially available virus (Takara Bio USA,

Mountain View, CA). RAW cells were grown in αMEM containing antibiotics and 10% FBS until 60% confluency. The ihMSCs were added to the cultures at the appropriate number and co-cultures were incubated for 18 h while stimulated with 0.5 µg mL$^{-1}$ LPS (Sigma). ELISA for tumor necrosis factor alpha (TNFα) was performed on recovered media (R&D Systems, Minneapolis, MN). Cells were recovered with trypsin/EDTA and transferred to a black 96-well costar plate (Corning, Corning NY). Fluorescence for GFP was read at 485/520 nm excitation/emission using a FluoStar plate reader (BMG Biotech, Ortenberg, Germany). A standard curve was used to determine RAW cell number and TNFα measurements were expressed normalized to this value.

**Mineralizing osteogenic differentiation assay.** Osteogenic differentiation and Alizarin Red S (ARS) staining was carried out using standard methods[27]. All reagents were obtained from Sigma unless stated otherwise. Once 90% confluency was reached, MSCs were incubated with CCM supplemented with 100 nM dexamethasone, 50 µg mL$^{-1}$ ascorbic acid, and 5 mM β-glycerophosphate for up to 21 days with new media added every 2 days to promote osteogenic differentiation and mineralization. At the appropriate time point, monolayers were washed twice with PBS and fixed with 10% (v/v) neutral-buffered formalin (VWR International, Radnor, PA) for 15 min. The fixed monolayers were washed with H$_2$O and then stained with 40 mM ARS pH 4.0 with orbital shaking at 100 r.p.m. for 30 min. The stain was removed and the monolayers were washed with distilled water. Micrographs were taken using an inverted microscope (Nikon Eclipse, TE200) fitted with a Nikon DXM1200F digital camera. Quantification of staining was performed by extraction and spectrophotometry as previously described[27]. Briefly, ARS attached to monolayer was dissolved in 10% (v/v) acetic acid prior to spectrophotometric quantification at 405 nm.

**Adipogenic differentiation assay.** Adipogenic differentiation and Oil Red O (ORO) staining were carried out in the standard manner[20,63]. All reagents were obtained from Sigma unless stated otherwise. Briefly, confluent monolayers of MSCs were incubated in CCM supplemented with 500 nM dexamethasone, 50 nM isobutylmethylxanthine, and 500 nM indomethacin for 21 days with changes of media every 2 days. On day 21, monolayers were washed with PBS and fixed with 10% (v/v) neutral-buffered formalin for 15 min. Fixed monolayers were then stained with 0.5% (w/v) Oil Red O dissolved in 50% isopropanol for 20 min. The monolayers were then washed with PBS to remove excess stain. Micrographs were taken using an inverted microscope (Nikon Eclipse, TE200) fitted with a Nikon DXM1200F digital camera. Quantification of staining was performed as previously described[65] by recovery of the stain with 50% isopropanol and spectrophotometric quantification at 405 nm.

**Chondrogenic differentiation assay.** All reagents were obtained from Sigma unless stated otherwise. One half-million MSCs were centrifuged at $500 \times g$ for 10 min to form a pellet. High-glucose Dulbecco's minimal-essential-media (Life Technologies) supplemented with 1 µM dexamethasone, 50 µg mL$^{-1}$ ascorbate-2-phosphate, 40 µg mL$^{-1}$ proline, 100 µg mL$^{-1}$ pyruvate, and 2× Insulin Transferrin Selenium-Plus Premix was added to the cell pellets for 21 days with fresh media added every 3 days to promote chondrogenic differentiation. On day 21, the pellets were washed with PBS and fixed with 10% (v/v) neutral buffered formalin for 15 min. The pellets were then embedded in paraffin and sectioned at 9 µm. Sections were stained with toluidine-borate solution to visualize sulfated proteoglycans. Micrographs were taken using an inverted microscope (Nikon Eclipse, TE200) fitted with a Nikon DXM1200F digital camera.

**Cell quantification by fluorescent dye intercalation (CyQuant).** Cell quantification was carried out using standard protocols[66]. Monolayers were frozen at −20 °C for 24 h before lysis buffer consisting of PBS containing 1 mM MgCl$_2$, 0.1% Triton X-100, and restriction enzyme 1 U mL$^{-1}$ EcoR1 and HindIII (Invitrogen) was added. Cells were incubated for 16 h in a humidified incubator at 37 °C with rocking. Samples and known standards were prepared in parallel. Lysate supernatants were recovered and transferred to a black 96-well plate. FluoStar plate reader (BMG Biotech) was used to read fluorescence at 485/520 nm excitation/emission. A standard curve was used to determine cell number.

**Quantitative reverse transcriptase PCR (qRT-PCR) assays.** Total RNA was extracted from the cells using a High Pure RNA isolation kit (Roche Diagnostics). Copy DNA was synthesized using a Superscript III kit (Life Technologies). The use of a random hexamer/oligo-dT combination was the only deviation from the manufacturer's protocol. SYBR Green or TaqMan gene expression assays (Applied Biosystems) were used to carry out qRT-PCR on a thermocycler fitted with a real-time module (CFX96; Biorad Laboratories, Hercules, CA). Fold changes were calculated using the $2^{-\Delta\Delta CT}$ method[67]. Primer sequences are provided in Supplementary Table S1. To generate heat maps, fold-change qRT-PCR data were log-transformed and z-normalized to the same scale, then plotted using Rstudio (v1.1.435), ggplot (3.0.0), dplyr (0.7.6), and reshape (1.4.3) programs.

**Immunoblotting.** MSC cultures were incubated in CCM (standard) or OBM (osteogenic) containing 10 µM GW9662 or vehicle (dimethyl sulfoxide) for up to 10 days prior to recovery by trypsin/EDTA and washing in cold PBS. Detergent-mediated soluble and insoluble fractionation was performed on the cells as through a modification of the protocol previously described by Ko et al.[65,68]. Briefly, cells were gently lysed by incubation in ice-cold 1% (v/v) Triton X-100 (Sigma) containing protease inhibitors (Roche Diagnostics, Indianapolis, IN) for 15 min followed by centrifugation at $14,000 \times g$ for 15 min. The supernatant was recovered (soluble fraction) and the pellet washed twice in excess lysis solution before resuspension in lysis solution containing 1% (w/v) sodium dodecyl sulfate. Immunoblotting was performed in the standard manner using Novex 4–20% tris-glycine gradient gels and associated reagents (Thermo Fisher). Blots were probed using mouse anti-human GAPDH (clone 6C5; Chemicon International, Temecula, CA #MAB374) at 1:1000, mouse anti-human β-catenin (clone 5H10; Chemicon #MAB2081) at 1:1000, mouse anti-human GSK3β (clone 3D10; Abcam, Cambridge, UK #ab93926) at 1:500, mouse anti-human PPARγ (clone 1E6A1; Thermo Fisher, #MA5-15417) at 1:500. Secondary antibody was goat anti-mouse Ig-peroxidase conjugate (cat#G210-040) (Thermo Fisher) at 1:1500. Signal development was carried out using hydrogen peroxide, luminol, and paracoumaric acid as previously described[69] using a Versadoc gel imager (Biorad, Hercules, CA). Densitometry was performed using Quantity One software (Biorad). Immunoblotting for collagen VI and XII was performed on whole-cell lysates using rabbit anti-human type VI collagen (NBP159126; Novus Biologicals, Littleton, CO) at 1:500, rabbit anti-human type XII collagen (NBP1-88062, Novus) at 1:500, goat anti-rabbit IgG-peroxidase conjugate (sc-2004, Santa Cruz Biotechnology, Dallas, TX) at 1:1500.

**ALP assays.** ALP assays were performed using standard protocols[20,66]. Briefly, MSCs were cultured up to 8 days in 12-well plates (Corning) in CCM or in the presence of osteogenic base media (OBM) consisting of CCM containing 5 mM β-glycerophosphate and 50 µg mL-1 ascorbate. On the day of measurement, the monolayers were washed twice with PBS and once with ALP reaction buffer (100 mM Tris-HCl, pH 9, 100 mM KCl, and 1 mM MgCl$_2$). Five hundred microliters of ALP reaction buffer was added to each well followed immediately by 500 µL p-nitrophenyl phosphate (PNPP; Life Technologies). The absorbance at 405 nm was recorded every 30 s for 10 min using a FluoStar plate reader (BMG Biotech). Results were normalized to cell number using a CyQuant fluorescent intercalation assay.

**ECM production.** ECM (also referred to as hOCM and ihOCM) was purified by a series of digestions and washes[20,70]. Reagents were obtained from Sigma unless stated otherwise. MSCs were grown to 70–80% confluency on a 154 mm$^2$ plate (Corning) in CCM before receiving OBM with either 10 µM GW9662 or an equal volume of DMSO vehicle. Media were replaced every 2 days for 10 days. On day 10, the monolayers were washed twice with PBS and frozen at −80 °C for 15 h. The monolayers were then thawed in the presence of room temperature PBS and scraped from the plate. The ECM/cell slurry was then pelleted by centrifugation at $1000 \times g$ for 10 min and resuspended in lysis buffer containing 0.1% (v/v) Triton X-100, 1 mM MgCl$_2$, and 10 µg mL$^{-1}$ DNAse I (Sigma). The ECM was incubated at 37 °C with orbital mixing at 60 r.p.m. for 2 h before addition of 0.01% (v/v) trypsin and incubation was continued for 15 h. The ECM was then washed twice in excess dH$_2$O, once in chloroform, once again in dH$_2$O, and once in acetone before being allowed to air dry. Dry ECM was stored at −80 °C.

**Protein, calcium, and glycosaminoglycan quantification.** A Pierce BCA protein assay kit (Thermo Fisher) was used following the manufacturer's instructions. Calcium quantification was performed on HCl-extracted samples using Arsenazo III reagent[27]. Briefly, samples were completely dissolved by boiling with reflux in 1 M HCl, then neutralized to pH 7.0 by addition of 1 M NaOH. Neutralized samples were assayed by addition of one volume of 100 µM Arsenazo III (Sigma) and spectrophotometric reading at 595 nm. GAG was assayed on matrix extracted from 152 cm$^2$ monolayers using a commercially available kit (Blyscan, Biocolor Life Science Assays, County Antrim, UK).

**Mass spectrometry and proteomic analysis.** Reagents were acquired from Fluka Honeywell Research Chemicals (Charlotte, NC) or Sigma unless otherwise stated. Samples were separated on a 4–20% Mini-Protean TGX precast gel (BioRad) and stained with colloidal Coomassie Blue (Sigma). Resultant bands were excised and de-stained by two cycles of 2:1 acetonitrile in 50 mM ammonium bicarbonate for 30 min, followed by 25 mM ammonium bicarbonate for 30 min. Proteins were reduced by 1 h incubation in 5 mM dithiothreitol following alkylation with 5.5 mM iodoacetamide. The bands were digested with 25 µg mL$^{-1}$ trypsin overnight at 36 °C. The sample was desalted and purified using a Ziptip C18 column according to the manufacturer's instructions (EMD Millipore, Burlington, MA). Digested samples were suspended in 0.1% formic acid (FA) before separation and analysis by LC-MS (Dionex UltiMate 3000 ultra-high performance liquid chromatography (UHPLC) system—Orbitrap Fusion Tribrid Mass Spectrometer, Thermo Fisher). The LC gradient was set up at 4 µl min$^{-1}$ using a 2–90% gradient of 0.1% (v/v) FA in acetonitrile (buffer B) against 0.1% (v/v) FA in dH$_2$O (buffer A). Detection was

performed with spray voltage of 2.3 KV, orbitrab resolution of 120 K, scan range of 400–1600, and higher-energy C-trap dissociation energy (HCD) set to 28%. Data were analyzed using Proteome Discoverer 2 software (Thermo). Clustering analysis was performed based on Euclidean distances in Rstudio (v1.1.435). Proteomic data sets accessible via the PRIDE Database (http://www.ebi.ac.uk/pride). Data are available via ProteomeXchange with identifier PXD016017.

**Scanning electron microscopy**. Dry ECM was iridium sputter coated with a Cressington 208HR high-resolution sputter coater (Cressington). Samples were imaged and EDS was performed using a FEI-Quanta 600 FE-SEM (Thermo Fisher Scientific).

**Transduction of hMSCs with lentiviral constructs expressing shRNA**. Human MSCs were expanded on 25 cm$^2$ monolayers in CCM until at a density approximately 5000 cells per cm$^2$. Cells were transduced at a multiplicity of infection of 10 with lentiviral particles expressing shRNA directed against the collagen VIα3 transcript (sc-94560-V, Santa Cruz), the collagen XIIα1 transcript (sc-72958-V, Santa Cruz) or non-targeting (scrambled) control shRNA (sc-108080, Santa Cruz) for 24 h in the presence of 9 μg mL$^{-1}$ polybrene (Sigma). After 24 h, monolayers were washed in PBS and cultured for a further 24 h before selection in CCM containing 2 μg mL$^{-1}$ puromycin (Sigma) for 2 days.

**Apoptosis assays**. Passage 4 scrambled, 6KD, and 12KD hMSCs were seeded onto 12-well plates at an initial seeding density of 2000 cells per cm$^2$ and allowed to grow as monolayers with changes of media every 2 days. At the appropriate time point, plates were washed with PBS and then stained with 2 μM Calcein AM (Biotium, Freemont, CA) in PBS for 45 min in the dark. Bottom fluorescence was obtained using a fluorescence plate reader (Victor Nivo, Perkin Elmer) in PBS (490/520 ex/em). Wells were then washed once with PBS and twice with 500 μM Annexin V-binding buffer (HEPES-buffered saline containing 2.5 mM CaCl$_2$) before staining with 0.75 μg mL$^{-1}$ Annexin V dye conjugate (Biotium CF®594) for 30 min in the dark. Wells were washed twice with 500 μL Annexin V-binding buffer and bottom fluorescence was obtained in Annexin V-binding buffer (561/594 ex/em). Wells were washed once with PBS and then fixed with 1 mL 4% (v/v) paraformaldehyde for 30 min in the dark. A blank plate was prepared by incubating a 12-well plate with 2 mL CCM containing 20% FBS per well for 4 h at 37 °C before staining.

**Immunocytochemistry for type VI and type XII collagen**. Immunocytochemistry was performed on monolayers at a density of about 10,000 cells per cm$^2$ on tissue culture treated plastic chamber slides. Monolayers were fixed for 5 min with phosphate-buffered paraformaldehyde, blocked, and permeabilized in PBS containing 0.1% Triton X-100 (Sigma) and 0.1% (v/v) goat serum (Sigma) then incubated with either rabbit anti-collagen XII (NBP1-88062, Novus) at 1:500 and/or mouse anti-collagen VI (MAB3303, EMD Millipore, MA) at 1:500. Mouse and rabbit antibodies were detected by Alexafluor- 488 (green) or Alexafluor 598 (red)-conjugated secondary antibodies, respectively (Invitrogen) at 1:500 dilution. Slides were processed with 4′,6-diamidino-2-phenylindole (DAPI) containing mount solution (Vector Laboratories, Burlingame, CA). Micrographs were captured with a Nikon Eclipse Ti microscope and processed with NIS Elements 4.2 software.

**Calvarial defect and cell administration**. Vertebrate animal studies were approved by the Institutional Animal Care and Use Committee and Texas A&M. Eight-week-old NU/NU mice were obtained from Charles River. A unilateral 4-mm-diameter full-thickness circular defect was generated in the parietal calvaria, approximately 2 mm from the sagittal and coronal sutures, using a 2.33 mm osteotomy burr (Roboz Surgical, Gaithersburg, MD). The defect is critical-sized and will not heal without intervention[28]. Two million MSCs or 5 × 10$^6$ human BM cells were suspended in 30 μL of reconstituted human plasma (Sigma) and 30 μL of thromboplastin (Plastinex, Horsham, PA) was then added before application to the defect. Before gelling of the plasma/cell solution, matrices, hOCM, ihOCM, or gelatin foam (Gelfoam, Baxter International, Deerfield, IL) were then administered to fill the defect. In some cases, gelatin foam saturated with 0.1 mg mL$^{-1}$ of BMP2 (Infuse, Medtronic, Minneapolis, MN) was applied to the plugged defect to serve as a control. The scalp was then sutured closed and analgesia was administered subcutaneously for 2 days after surgery.

**Calvaria excision and processing**. Following euthanasia, the calvaria were cut from the skull with a 5 mm diameter fine rotary blade (Strauss Diamond, Palm Coast, FL), washed in PBS, and placed in 10% neutral-buffered formalin. After 24 h calvaria were washed in PBS, transferred to Carson's fixative (1.86% (w/v) sodium phosphate monobasic, 0.42% (w/v) sodium hydroxide, 10% (v/v) formaldehyde), and stored at 4 °C.

**μCT analysis**. Samples were removed from Carson's fixative, washed in PBS, and wrapped in Parafilm (VWR International) immediately prior to μCT analysis. A Skyscan 1275 micro-computed-tomography scanning system (Bruker) was used for μCT scans. Calvaria were scanned over 360° using a 30 kV beam with a camera resolution of 18 μM, flat field correction and frame averaging were enabled. Images were captured every 0.5°. Axial reconstructions were generated using NRecon (Micro Photonics, Allentown, PA). Smoothing and beam hardening were kept consistent throughout the study at 1% (smoothing kernel gaussian) and 5%, respectively. Misalignment compensation, ring artifact reduction, and cross-sectional rotation were adjusted as needed to minimize scan artifacts. The dynamic range was kept between 0 and 0.111204 for all reconstructions. Mock control specimens consisting of calvarial bones with defects plugged with fresh ihOCM were employed to optimize beam intensity and to determine the specific radio-densities of bone and ihOCM. A threshold was then determined that removed all of the ihOCM density from the reconstructions. This threshold was applied to the samples in CTvox and CTan to distinguish between soft ECM and new bone in the defect. Comparison with histological sections later validated this approach. Calcium hydroxyapatite phantoms (Bruker) were employed to calibrate the instrument and facilitate determination of BMD using the attenuation coefficient method. For determination of BMD on other tissues, bones from the mice used in the healing study were utilized. For the spine, the last vertebra was selected, specifically a 0.18 mm$^3$ region of interest of cortical bone located on the neural arch of the vertebrae. For the pelvis, a 0.18 mm$^3$ region of cortical bone located on the cranial aspect of the iliac crest was used. For the femur a 0.18 mm$^3$ region of interest of cortical bone on the distal aspect of the third trochater was used for BMD measurements. After the μCT scans were complete, the data set was reconstructed using the NRecon protocol. The reconstructed data set was then used to make a 2D image, as well as determine the 3D surface to volume ratio, utilizing the DataViewer protocol. In accordance with the Bruker microCT "Bone mineral density (BMD) and tissue mineral density (TMD) calibration and measurement using Bruker microCT BMD phantoms and CT-Analyser software" protocol, the calcium hydroxyapatite phantoms (Bruker) were employed to calibrate the instrument and facilitate determination of BMD using the attenuation coefficient method prior to determining the BMD of each of the samples. CTvox software (Micro Photonics) was used to generate three-dimensional models from the reconstructions and CTan software was used to perform measurements on reconstructions in accordance with the manufacturer's (Micro Photonics) instructions.

**Histology**. Calvaria were removed from Carson's fixative, washed with PBS, and transferred to 1 M dibasic EDTA, pH 8.0 (Sigma) for decalcification. The solution was changed every 2 days and radiolucency was confirmed by scanning. Following decalcification, the samples were dehydrated through increasing gradients of alcohols, cleared with Sub-X clearing agent (Surgipath Medical Industries, Richmond, IL), and embedded in paraffin (Richard-Allan Scientific, San Diego, CA). Paraffin-embedded samples were cut to 9 μm sections and floated onto Superfrost Plus microscope slides (Fisher Scientific). Prior to staining, sections were baked onto the slides at 60 °C for 1 h, deparaffinized with Sub-X, and rehydrated. For H&E staining, sections were stained in hematoxylin solution Gill number 3 (Sigma) and counterstained with 1% (w/v) eosin Y (Thermo Fisher) before clearing and dehydration. Masson's trichrome staining was achieved using a commercially available kit (American Master Tech, Lodi, CA) following the manufacturer's instructions. For TRAP staining, an acid phosphatase/leukocyte kit (Sigma) was used according to the manufacturer's instructions. TRAP staining was quantified using ImageJ (NIH). Slides were coverslipped using Permount mounting medium (Fisher Scientific).

**Statistics**. GraphPad Prism version 5.00 for windows was used to plot data and carry out statistical analysis. Single means were compared using *t*-tests while multiple tests of means were carried out using one-way analysis of variance (ANOVA) and either Dunnett's or Tukey's post-test where appropriate. Specific statistical parameters are given in the figure legends.

**Reporting summary**. Further information on research design is available in the Nature Research Reporting Summary linked to this article.

## Data availability

Proteomic data sets accessible via the PRIDE Database (http://www.ebi.ac.uk/pride) accession number: PXD016017. Complete blots and raw numerical data are available in a supplemental Source Data file. Source data are provided with this paper.

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

## Acknowledgements

For scanning electron microscopy the use of the Texas A&M Microscopy and Imaging Center and Mr. Tom Stephens are acknowledged. For mass spectrometry the use of the Texas A&M University Laboratory for Biological Mass Spectrometry and Dr. Doyong Kim are acknowledged. This work was funded by research grants from the National Institute of Arthritis and Musculoskeletal and Skin Diseases R01AR066033 and R21AR072292A1, an Investigator Initiated Award from the Cancer Prevention Research Institute of Texas and an X-grant from the Texas A&M President's Excellence Fund.

## Author contributions

C.A.G., E.P.M., and R.K conceived the study. C.A.G., R.K., E.P.M., S.Z., S.P., A.H., M.C., B.H.C., U.K., L.K.D., M.G., C.K., D.T., and W.B.S. performed experiments and interpreted data. F.L. and Q.Z. generated and provided the ihMSCs. E.P.M., S.Z., C.A.G., and R.K. wrote the manuscript. All authors proofed and approved the manuscript.

## Competing interests

The authors declare no competing interests.
