## [Peer Review File · Nature Communications]

Reviewers' Comments:

Reviewer #1:

Remarks to the Author:

The manuscript by McNeill et al. derives an osteogenic cell-matrix and tests its bone anabolic activity in a calvarial defect model in mice. The premise of the study is well-justified as few therapeutics exist that exhibit anabolic activity in bone. The study design is also rigorous, and the presented data is of high quality. The statistical analyses appear appropriate and methods are sufficient to allow the work to be reproduced. However, several concerns were identified related to specific methods and data sets that should be addressed prior to publication.

Major concerns

1. The authors present a rigorous set of data characterizing the tri-lineage potential of ihMSCs and BM-MSCs. However, the desire to characterize their immuno-suppressive and anti-inflammatory activities is questionable given the premise of the study, and the data presented in this regard are modest at best. Therefore, it would seem prudent to relegate these data (Figure 1e, f) to a supplementary figure. One concern is that CFSE-based labeling shows a complete inhibition of cell proliferation by ihMSCs and a loss of label in control cells, which is atypical. The authors should reduce the incubation time/CFSE labeling to fall within the dynamic range of the assay and include different ratios of effector to target cells. BM-MSCs and an unmixed PBMC control to determine the degree of stimulation should also be included.
2. The authors demonstrate that OEihMSCs generate higher yields of ECM as compared to OEhMSCs, which is enhanced by GW9662, without altering protein content. Assuming the increased mass represents GAGs, it would be useful to demonstrate this by quantitative analysis.
3. Data in Figure 3a,b is curious in that OEihMSC and OEhMSC1 cluster together but GW9662 treatment increases the distance between these samples. This likely reflects the fact that GW9662 treatment reduces to zero the number of ECM proteins shared by these two populations (Fig. 3a). GW9662 treatment increases the number from 1 to 3 of proteins unique to OEihMSCs. What are these proteins and were they tested for their osteogenic promoting activity?
4. The authors incubate that "treated ihOCM upregulated the activity of ALP and the early osteogenic biomarker osteoprotegerin (OPG) by attached ihMSCs" but the legend to Figure 3g states that "hMSCs were attached to ihOCM generated in the presence of GW9662 (grey) or vehicle (white)". Please clarify if ihMSCs or hMSCs were used. In either case, it would seem appropriate to test both matrices and populations in combination to validate the superior activity of ihOCM in this assay. Does the matrix induce any activity in the absence of osteogenic stimulants?
5. Data based on knockdown studies appear rigorous, but it may be useful to evaluate effects on apoptosis/viability, e.g. do cells grow more slowly or is their rate of cell death higher? This may be important in evaluating the mineralization data from Figure 4g, e.g. why does mineralization fall off between D14 and D21 in the KD12 sample.
6. All studies evaluate early markers of mineralization, such as ALP. Is it valid to look at markers of secondary mineralization (late stage), such as osteocalcin, Mepe, etc.?
7. Results of calvarial defect model studies are compelling. Did the authors evaluate control mice with defects for longer time periods to ensure the 4mm hole represents a critical size defect, e.g. no significant healing? Effects of the ihmatrix (assume this is equivalent to ihOCM) is remarkable and inclusion of an additional timepoint may be beneficial to demonstrate complete healing vs. other formulation tested.
8. How was healing index calculated? Also, since authors are proposing an osteogenic matrix to accelerate fracture healing, they should provide a rationale for using the calvarial defect model

(non-weight bearing bone) vs. a tibia burr hole defect model (weight bearing). Ultimately, the matrix should be tested in a surgical defect model to evaluate the integrity/strength of the repaired bone.

Minor comments:

1. In Suppl Figure 2b, graphs illustrating results of qPCR analyses employ different scales, which makes evaluation of differences between groups difficult. It is suggested the authors use a single scale, which would magnify differences in gene induction among the different cells tested.
2. Data in Suppl Figure 3a demonstrating effects of troglitazone or CCT032374 alone or in combination on ihMSC adipogenic is qualitative and therefore subjective. Quantitative analysis should be performed to assess stimulatory effect of each agent and if they work additively in combination.
3. In Figure 2b, PPARgamma2 protein abundance is normalized to GAPDH, but the control used in the blot is beta-actin. Same error appears in Figure 2d with beta-catenin. Please clarify.
4. Is ihmatrix equivalent to ihOCM? Please keep terminology consistent as it is confusing enough.

Reviewer #2:

Remarks to the Author:

This manuscript reports on a study in which hMSC were derived from hiPSC, and were then used to generate an extracellular matrix (ihOCM). This matrix and the cells that produced it were carefully characterized and were compared to bone marrow-derived MSC and the matrix that they produce. The role of Wnt signaling and PPARγ in the osteogenic capacity of the ihMSC was investigated, as was the role of Col VI and Col XII proteins, which were found to be more abundant in ihOCM than in MSC-produced matrix. The effect of ihOCM on bone regeneration in a calvarial defect in the nude mouse was then assessed, relative to controls, and with/without the addition of cells. The main findings of the paper relate to the composition and osteogenic properties of the ihOCM, with particular emphasis on the roles of collagen VI and XII. The authors assert that "Based on these findings, we propose that ihOCM may represent an effective replacement for the current gold standards, autograft and BMP products, used commonly in bone tissue engineering."

This paper is of interest to those in the regenerative medicine community who study the biology and application of progenitor cells, in particular those interested in the application of cell and biomaterials to bone healing. The manuscript is generally well prepared, the data are for the most part clearly presented, and the study is technically sound. The work on characterizing the ihMSC and ihOCM is particularly comprehensive. However there are several issues with this paper in relation to its place in the field:

1] The use of cell-derived extracellular matrix to guide cell function and to ultimately potentiate regeneration has been studied widely. More specifically, the use of MSC-derived matrix to promote osteogenesis has been previously reported by several groups, including in previous publications by the authors (Refs 20-22). The main innovation in this paper seems to be the use of iPSC-derived MSC, the derivation of which is also widely reported in the literature, for similar purposes. The impact of this paper on the field of bone regeneration is therefore not evident, and it would be helpful to explain how this study significantly extends the preceding work by a number of groups internationally.

2] One aspect of this study's approach that is highlighted is that the use of iPSC as "a theoretically limitless and reproducible source of cells" (p. 3). However, it is now established that iPSC from

different donors can exhibit practically significant variability (see e.g. Nature 2017, 546:370-375) and furthermore that derivation of iPSC is not always consistent. For a proposed patient-specific therapeutic approach, this would seem to be problematic. The reported study uses "a single line of iPSCs" (p. 11), which calls into question the broader applicability of the approach, in particular since the authors clearly point out the inter-donor variability of MSC. The study would be improved by validating that iPSC-derived MSC are in fact more consistent across donors, as this seems to be a key to this approach being an improvement.

3] The data in the paper focus largely on the properties of the ihMSC and the composition of the resulting ihOCM, including the molecules and mechanisms that are involved in osteogenesis, which appear to be the same as in bone marrow-derived MSC. These in vitro studies are quite comprehensive and well done, and provide good evidence for the conclusions the authors draw regarding the in vitro characterization of the cells and matrix. However, the title of the paper highlights the "osteoregenerative capabilities" of the ihOCM, and relatively few data on bone regeneration are reported; of the 43 panels of data presented, only 7 contain data on bone regeneration, and these are not very comprehensive. The paper's main point appears to be that iPSC-derived MSC are similar to other MSC in terms of the components and mechanisms by which they promote osteogenesis, and may be more potent (if it can be shown that other/all ihMSC behave similarly). This is an interesting finding, though not one that is reflected in the title of the paper.

4] The manuscript is generally very well-written and clear, and the Methods are adequately described; the exception is the derivation of the iPSC, which is important but included only by reference to a previous paper (which is incorrectly cited; it is presumably Ref 17). The Discussion section is quite repetitive of the Results section, and could be made more concise by focusing on the impact of the findings and their relevance to the current state of the field. The final paragraph of the Discussion is not compelling in terms of describing how these findings will influence the thinking of others in the field. There are a few awkward phrasings and terminology; e.g. use of the term "holy grail" is rather colloquial for a scientific manuscript, the authors refer to "stress-processes" (perhaps they mean "stress fibers"?), there is a typo in the legend to Figure 2.

We appreciate the considerable amount of time the reviewers have invested in comprehensively reviewing our manuscript. The critiques are insightful, fair, and we have endeavored to respond to each of them with manuscript revisions and/or additional data. As per editorial request, we have highlighted changes in the revised text, and indicated their positions in the responses below.

Reviewer #1:

Major concerns.

1. The authors present a rigorous set of data characterizing the tri-lineage potential of ihMSCs and BM-MSCs. However, the desire to characterize their immuno-suppressive and anti-inflammatory activities is questionable given the premise of the study, and the data presented in this regard are modest at best. Therefore, it would seem prudent to relegate these data (Figure 1e, f) to a supplementary figure. One concern is that CFSE-based labeling shows a complete inhibition of cell proliferation by ihMSCs and a loss of label in control cells, which is atypical. The authors should reduce the incubation time/CFSE labeling to fall within the dynamic range of the assay and include different ratios of effector to target cells. BM-MSCs and an unmixed PBMC control to determine the degree of stimulation should also be included.

In response to this concern we have placed the immune modulation data in a supplemental figure (**new Figure S1, text description page 5**) with the desired controls. We have also included an additional condition where a 1:100 ratio of MSCs to PBCs is assayed. This condition allows better resolution of the individual populations that are detected during dilution of CFSA and is within the dynamic range of the assay. For comparison, we have also included CFSE data for bone marrow derived hMSC (donor hMSC-1).

2. The authors demonstrate that OEihMSCs generate higher yields of ECM as compared to OEhMSCs, which is enhanced by GW9662, without altering protein content. Assuming the increased mass represents GAGs, it would be useful to demonstrate this by quantitative analysis.

It is probable that the increased yield with GW9662 treated ihOCM is due to increased protein production because the values in Fig2h are normalized to overall mass, and protein composition by mass is not significantly altered between the groups. Nevertheless, after reading your question we got curious and performed the GAG measurements for GW9662 generated matrices. GAG values were calculated and normalized to yield mass and we found that the ratios of GAG to protein were comparable between groups.

Figure 2 has been modified to incorporate the new data. **Text page 7.**

3. Data in Figure 3a,b is curious in that OEihMSC and OEhMSC1 cluster together but GW9662 treatment increases the distance between these samples. This likely reflects the fact that GW9662 treatment reduces to zero the number of ECM proteins shared by these two populations (Fig. 3a). GW9662 treatment increases the number from 1 to 3 of proteins unique to OEihMSCs. What are these proteins and were they tested for their osteogenic promoting activity?

This is an astute observation. Note that upon exposure to GW9662, the number of ECM proteins shared by 2 of the MSC samples (hMSC-1, hMSC-2 and ihMSCs) is always reduced and the proteins shared by all 3 are increased from 16 to 25. Upon inspection of the list, it appears that GW9662 generalizes the composition of the OCM increasing the number of components shared by all 3 rather than only 2. This is particularly critical for the least osteogenic (hMSC2) donor which loses protein specifically to itself and gains osteogenic proteins shared by the other 2 more osteogenic MSC samples (hMSC1 and ihMSC). We have endeavored to clarify this complex interpretation in the **Results, page 7**).

As we have demonstrated, the function of the OCM from all MSC-sources can be attributed to the action of collagen VI and XII, but this does not fully explain the superiority of the ihOCM. We agree that the presence of a unique protein in ihMSCs, after treatment of GW9662, could represent a potential explanation for increased osteogenic efficacy. The protein signatures unique to ihMSCs upon GW9662 exposure are fibulin-2, the alpha-5 chain of collagen IV and the alpha chain of type VIII collagen. Of the three, collagen IV and VIII are present in GW9662-untreated matrices, effectively excluding them as a candidate for the superior osteogenic activity of GW9662-treated ihOCM. The presence or action of fibulin-2 in developing bone tissue is not well documented in the literature, but its presence raises the attractive possibility of a mechanism where osteogenic BMP2 is concentrated and its activity potentiated by strong interactions with collagen VI (1, 2), and the complex is tethered to the ECM via fibulin-2 and its dual affinities for collagen IV and collagen VI (3). This hypothesis is raised in the **Discussion, page 14** with the supporting references cited above.

4. The authors incubate that “treated ihOCM upregulated the activity of ALP and the early osteogenic biomarker osteoprotegerin (OPG) by attached ihMSCs” but the legend to Figure 3g states that “hMSCs were attached to ihOCM generated in the presence of GW9662 (grey) or vehicle (white)”. Please clarify if ihMSCs or hMSCs were used. In either case, it would seem appropriate to test both matrices and populations in combination to validate the superior activity of ihOCM in this assay. Does the matrix induce any activity in the absence of osteogenic stimulants?

We tested ihMSCs and hMSC-1 on the ihOCM preparation with virtually identical outcomes. The data for ihMSCs are presented in Fig3. We don't observe significant osteogenic activity in the absence of supplementation with beta glycerophosphate/ascorbic acid. Changes made to reflect these clarifications are provided in the **Results, page 8** and in the **figure legend of Fig3**.

5. Data based on knockdown studies appear rigorous, but it may be useful to evaluate effects on apoptosis/viability, e.g. do cells grow more slowly or is their rate of cell death higher? This may be important in evaluating the mineralization data from Figure 4g, e.g. why does mineralization fall off between D14 and D21 in the KD12 sample.

This is an interesting question, and while we did not see particular signs of excess apoptosis in the knock down MSCs even though there was a subtle morphological change, we agree that measurements are warranted. We have performed extra experiments on the collagen VI and XII knock down cells to measure apoptosis in the growing cultures. We utilized an assay that measured exposed annexin V binding while normalizing the live cells in monolayers to calcein AM staining. We found that overall, apoptosis levels were very low, but peaked a little when the rate of cell expansion increased or decreased through the lag-log transition or log-plateau transition. Apoptosis was not increased by knock down of either collagen, and surprisingly, the highest transient apoptosis levels were seen in the controls (presumably due to larger differences in the rates of growth at transitional stages of the growth curve). These results are provided in the supplemental data. New **Figure S8 Panel c**, description in **text page 8**.

6. All studies evaluate early markers of mineralization, such as ALP. Is it valid to look at markers of secondary mineralization (late stage), such as osteocalcin, Mepe, etc.?

A good question. To assess the late stages of osteogenesis, mineralization assays are provided for characterization of the ihMSCs and hMSC donors (Fig1g,h), for characterization of the effects of GW9662 (Fig2f), and also for the knock-down collagen

VI and XII studies (Fig4g). However, assaying late stage osteogenic biomarkers expressed by cells cultured on OCM is complicated by the fact that they coat the monolayer with their own ECM during culture. The amount of OCM that is coated onto the plastic is blocked by cell-derived ECM in a matter of days thereby perturbing results. For assays that test the osteogenic efficacy of coated OCM, we are therefore restricted to early markers such as OPG or mid-stage markers such as ALP. A clarification is added to **Results, page 8**.

7. Results of calvarial defect model studies are compelling. Did the authors evaluate control mice with defects for longer time periods to ensure the 4mm hole represents a critical size defect, e.g. no significant healing?

In our previous publication (Krause et al. 2010 (4)), we confirmed that the defects are critical up to 50 days. Clarification is made in the **Methods Section, page 10** and **Discussion page 14**.

Effects of the ihmatrix (assume this is equivalent to ihOCM) is remarkable and inclusion of an additional timepoint may be beneficial to demonstrate complete healing vs. other formulation tested.

The calvarial defect assay employed a rapid 4-week endpoint which is already about half the duration usually employed for assays of this type (5). Given the difficulties we encountered detecting activity of the positive BMP2 control after 4 weeks, an intermediate time point would likely demonstrate healing in the ihOCM groups, and a small amount of activity in the cell-only groups (as we have demonstrated previously (6)) but we would be unable to compare them to the positive control. Respectfully, we believe that the data generated would have marginal additional impact on the current findings of the study, and it would take over 3 months to perform the experiments needed with the correct controls.

8. How was healing index calculated? Also, since authors are proposing an osteogenic matrix to accelerate fracture healing, they should provide a rationale for using the calvarial defect model (non-weight bearing bone) vs. a tibia burr hole defect model (weight bearing). Ultimately, the matrix should be tested in a surgical defect model to evaluate the integrity/strength of the repaired bone.

Agreed, but at the time of performing these experiments our main priority was assessment of basic osteogenic activity of the pure material rather than its performance in clinically-relevant weight-bearing models. In our hands, the calvarial defect is critical-sized, self-stabilizing, and requires a very small volume of material, thereby dismissing the need for fillers to satisfy volumetric deficits or scaffolds to satisfy stiffness requirements. Nevertheless, experiments with a long bone defect will need to be done, and we are well positioned to do this with our segmental femoral defect assay performed in nude mice when protocols for scaffold generation and crosslinking are perfected. **This is addressed in the Discussion, page 14**.

Testing the strength of the *de novo* bone in a calvarial defect is challenging, but we agree that some measure of strength would increase the impact of the work. Biomechanical testing is not possible at this stage because the original specimens were utilized for histology, but we have performed bone mineral density and surface:volume measurements on the *de novo* bone and compared it with weight-bearing bones such as cortical femur, lumbar vertebrae and pelvis from the same host animals. BMD and surface:volume are excellent surrogates for biomechanical strength testing (7, 8). These new data are presented in **FigS11c**, described in **Results, page 10** and **Discussion, page 15**.

Minor comments:

1. In Suppl Figure 2b, graphs illustrating results of qPCR analyses employ different scales, which makes evaluation of differences between groups difficult. It is suggested the authors use a single scale, which would magnify differences in gene induction among the different cells tested.

Data plotted in additional **FigS3c** to facilitate comparison of fold-changes between MSC preparations.

2. Data in Suppl Figure 3a demonstrating effects of troglitazone or CCT032374 alone or in combination on ihMSC adipogenic is qualitative and therefore subjective. Quantative analysis should be performed to asses stimulatory effect of each agent and if they work additively in combination.

There appears to be no additive or synergistic effects of the two drugs, but they are equally effective as individual agents. Data plotted in additional **FigS4b**.

3. In Figure 2b, PPARgamma2 protein abundance is normalized to GAPDH, but the control used in the blot is beta-actin. Same error appears in Figure 2d with beta-catenin. Please clarify.

Thank you for spotting this error. **Fig2b&d now corrected.**

4. Is ihmatrix equivalent to ihOCM? Please keep terminology consistent as it is confusing enough.

Thank you for spotting this error, now corrected throughout.

Reviewer #2:

1] The use of cell-derived extracellular matrix to guide cell function and to ultimately potentiate regeneration has been studied widely. More specifically, the use of MSC-derived matrix to promote osteogenesis has been previously reported by several groups, including in previous publications by the authors (Refs 20-22). The main innovation in this paper seems to be the use of iPSC-derived MSC, the derivation of which is also widely reported in the literature, for similar purposes. The impact of this paper on the field of bone regeneration is therefore not evident, and it would be helpful to explain how this study significantly extends the preceding work by a number of groups internationally.

The authors agree that MSCs derived from iPS cells have been described by several groups and their potential for the generation of bone tissue as a cytotherapeutic agent has been reported in several articles. Likewise, there are examples of the use of MSC-derived ECM for tissue engineering applications, including bone.

The overarching innovation herein lies in the combination of these individual approaches for the generation of osteogenic ECM in large batches from a reproducible source of iPS-derived MSCs. The goal of this approach is to reduce donor-batch variation and generate a material with predictable and reproducible composition and characteristics. To our knowledge (and based on extensive PubMed searches), this is the first comprehensive description of an iPS-MSC derived matrix with potent osteogenic and osteoregenerative properties and the first attempt to definitively identify the biologically active components of MSC-derived matrices. Moreover, this is the first report of an iPS-derived ECM (ihOCM) with intrinsic osteogenic activity in vivo. Unlike bone-marrow derived hOCM preparations and the vast majority of clinically employed bone scaffolds, the ihOCM material described here has a surprising level of efficacy, competing with the effectiveness of BMP2 under the experimental conditions we employed, dismissing need for added cells to boost activity.

To address concerns regarding the contribution of this work to the existing field, additions have been made to the **Discussion, page 16.**

2] One aspect of this study's approach that is highlighted is that the use of iPSC as "a theoretically limitless and reproducible source of cells" (p. 3). However, it is now established that iPSC from different donors can exhibit practically significant variability (see e.g. Nature 2017, 546:370-375) and furthermore that derivation of iPSC is not always consistent. For a proposed patient-specific therapeutic approach, this would seem to be problematic. The reported study uses "a single line of iPSCs" (p. 11), which calls into question the broader applicability of the approach, in particular since the authors clearly point out the inter-donor variability of MSC. The study would be improved by validating that iPSC-derived MSC are in fact more consistent across donors, as this seems to be a key to this approach being an improvement.

Agreed, iPSC cells from different sources do exhibit variation that seems to arise from the methodology and/or the tissue of origin. While the means to reproducibly generate ihOCM from every iPSC cell line (or even several) would be incredibly impactful, we simply did not have the means to incorporate this into the scope of the paper. Instead, the manuscript offers the means to generate ihOCM from one iPSC line with very reproducible composition. Because the final product is characterized and cell free, it could be utilized in the allogeneic setting without the concern of donor variation. We have attempted to address the issue of variation in iPSCs, and our rationale for focus on one iPSC line in the **Discussion, page 16.**

3] The data in the paper focus largely on the properties of the ihMSC and the composition of the resulting ihOCM, including the molecules and mechanisms that are involved in osteogenesis, which appear to be the same as in bone marrow-derived MSC. These in vitro studies are quite comprehensive and well done, and provide good evidence for the conclusions the authors draw regarding the in vitro characterization of the cells and matrix. However, the title of the paper highlights the "osteoregenerative capabilities" of the ihOCM, and relatively few data on bone regeneration are reported; of the 43 panels of data presented, only 7 contain data on bone regeneration, and these are not very comprehensive. The paper's main point appears to be that iPSC-derived MSC are similar to other MSC in terms of the components and mechanisms by which they promote osteogenesis, and may be more potent (if it can be shown that other/all ihMSC behave similarly). This is an interesting finding, though not one that is reflected in the title of the paper.

This is a point well taken. Our original goal was to generate a reproducibly manufacturable and highly osteogenic ECM for therapy in humans, but to do this in a meaningful, scientific and impactful way, a large body of mechanistic data had to be generated. In an attempt to preserve the translational impact of the manuscript, but communicate the other elements of the study, we propose the following title:

"A Pluripotent Stem Cell-Derived Matrix with Powerful Osteoregenerative Capabilities Shares the Basic Composition of Bone-Marrow Stromal Cell Matrices but is Superior in Efficacy."

4] The manuscript is generally very well-written and clear, and the Methods are adequately described; the exception is the derivation of the iPSC, which is important but included only by reference to a previous paper (which is incorrectly cited; it is presumably Ref 17). The Discussion section is quite repetitive of the Results section, and could be made more concise by focusing on the impact of the findings and their relevance to the current state of the field. The final paragraph of the Discussion is not compelling in terms of describing how these findings will influence the thinking of others in the field. There are a few awkward phrasings and terminology; e.g. use of the term "holy grail" is rather colloquial for a scientific manuscript,

the authors refer to “stress-processes” (perhaps they mean “stress fibers”?), there is a typo in the legend to Figure 2.

The Discussion has been revised in several areas to improve impact and describe the state of the field in more detail.

“A safe and manufacturable material that mimics anabolic bone is the holy grail of bone tissue engineering, but achieving this is challenging.” now reads “A safe and manufacturable material that mimics anabolic bone is the primary goal of bone tissue engineering, but achieving this is challenging.”

“...stress processes...” now reads “...cellular processes...”

Figure 2 now corrected.

References.

1. Paralkar VM, Weeks BS, Yu YM, Kleinman HK, Reddi AH. Recombinant human bone morphogenetic protein 2B stimulates PC12 cell differentiation: potentiation and binding to type IV collagen. *J Cell Biol.* 1992;119(6):1721-8. doi: 10.1083/jcb.119.6.1721. PubMed PMID: 1469059; PMCID: PMC2289768.
2. Gao T, Lindholm TS, Marttinen A, Urist MR. Composites of bone morphogenetic protein (BMP) and type IV collagen, coral-derived coral hydroxyapatite, and tricalcium phosphate ceramics. *Int Orthop.* 1996;20(5):321-5. doi: 10.1007/s002640050086. PubMed PMID: 8930726.
3. Sasaki T, Gohring W, Pan TC, Chu ML, Timpl R. Binding of mouse and human fibulin-2 to extracellular matrix ligands. *J Mol Biol.* 1995;254(5):892-9. doi: 10.1006/jmbi.1995.0664. PubMed PMID: 7500359.
4. Krause U, Harris S, Green A, Ylostalo J, Zeitouni S, Lee N, Gregory CA. Pharmaceutical modulation of canonical Wnt signaling in multipotent stromal cells for improved osteoinductive therapy. *Proc Natl Acad Sci U S A.* 2010;107(9):4147-52. Epub 2010/02/13. doi: 0914360107 [pii]10.1073/pnas.0914360107. PubMed PMID: 20150512; PMCID: 2840116.
5. Murphy MP, Quarto N, Longaker MT, Wan DC. (*) Calvarial Defects: Cell-Based Reconstructive Strategies in the Murine Model. *Tissue Eng Part C Methods.* 2017;23(12):971-81. doi: 10.1089/ten.TEC.2017.0230. PubMed PMID: 28825366; PMCID: PMC5734144.
6. Zeitouni S, Krause U, Clough BH, Halderman H, Falster A, Blalock DT, Chaput CD, Sampson HW, Gregory CA. Human mesenchymal stem cell-derived matrices for enhanced osteoregeneration. *Sci Transl Med.* 2012;4(132):132ra55. Epub 2012/05/04. doi: 4/132/132ra55 [pii]10.1126/scitranslmed.3003396. PubMed PMID: 22553253.
7. Bagi CM, Hanson N, Andresen C, Pero R, Lariviere R, Turner CH, Laib A. The use of micro-CT to evaluate cortical bone geometry and strength in nude rats: correlation with mechanical testing, pQCT and DXA. *Bone.* 2006;38(1):136-44. doi: 10.1016/j.bone.2005.07.028. PubMed PMID: 16301011.
8. Engelke K, Libanati C, Fuerst T, Zysset P, Genant HK. Advanced CT based in vivo methods for the assessment of bone density, structure, and strength. *Current osteoporosis reports.* 2013;11(3):246-55. doi: 10.1007/s11914-013-0147-2. PubMed PMID: 23712690.

Reviewers' Comments:

Reviewer #1:

Remarks to the Author:

The authors should be commended for their efforts to address concerns raised during the initial review of their manuscript. These concerns were all addressed in a systematic manner, and in many cases the response involved inclusion of new experimental data. Congratulations on an excellent study.